# FuseGPT: Prune-and-Fuse Knowledge Redistribution for Efficient Transformers

## Abstract

Structured pruning of Generative Pre-trained Transformers (GPTs) offers a promising path to efficient models, but often at the cost of performance degradation from discarded transformer blocks. In this paper, we introduce FuseGPT, a compression paradigm that reframes structured pruning as knowledge redistribution rather than simple removal. Instead of discarding less salient blocks, FuseGPT recycles them by fusing their knowledge into neighboring blocks, thereby preserving the model's performance. Our approach has two core components. First, we propose a fusion-aware importance metric, Macro Influence (MI), that identifies blocks not by their redundancy, but by their capacity to be effectively absorbed by other blocks. Second, we introduce a learnable layers fusion mechanism that uses low-rank matrices to graft the knowledge from a pruned block onto its neighbors. This process is guided by a lightweight, group-level fine-tuning procedure that uses a distillation-based loss to ensure the fused knowledge is properly integrated. FuseGPT works for both large language and multimodal models, generally surpassing representative prior methods in perplexity and zero-shot task performance, using as few as 32 calibration and 1024 fine-tuning samples. This "prune-and-fuse" approach opens a new avenue for model compression, focusing on repurposing rather than discarding valuable pre-trained knowledge.

## 1 Introduction

Generative Pre-trained Transformers (GPTs) have demonstrated remarkable capabilities in handling complex tasks and exhibiting emergent abilities in various domains, especially when scaled to billions of parameters Brown (2020); Zhang et al. (2022); Touvron et al. (2023); Liu et al. (2024c). Despite their unprecedented success, the increasing complexity and size of GPTs have introduced significant challenges for deployment in real-world scenarios, particularly in resource-constrained environments.

To address the hardware demands associated with deploying GPTs, model compression techniques are developed to produce more compact models while preserving high performance. These techniques primarily fall into two categories: model pruning and quantization LeCun et al. (1989); Han et al. (2015); Hoefler et al. (2021); Liu et al. (2021); Gholami et al. (2022). This paper focuses on model pruning, a technique aimed at reducing model size by eliminating redundant parameters. Pruning is mainly categorized into two types: unstructured pruning and structured pruning Frantar & Alistarh (2023a); Wang et al. (2019). Unstructured pruning targets removing individual weights, which can achieve higher performance but often results in hardware-unfriendly sparse weights, limiting acceleration potential. Structured pruning, in contrast, removes entire pre-defined structures (e.g., layers or blocks) at once, which may lead to a slight reduction in accuracy but is more hardware-efficient.

Recent studies have revealed that redundancy exists across transformer blocks in GPTs, meaning that certain blocks contribute less significantly to the final outcomes Men et al. (2024); Kim et al. (2024); Song et al. (2024). Some existing methods detect this redundancy by analyzing the similarities between hidden states, while others directly measure the changes in distance to the hard labels. Once redundant blocks are identified, structured pruning is applied to remove the least important ones, aiming to minimize the performance degradation. However, simply discarding these blocks often results in irreversible performance loss. While traditional post-pruning fine-tuning can help recover the performance, they typically require extensive datasets and substantial computational resources. As a result, there is a pressing need for more efficient methods to restore model performance without such heavy demands.

In this paper, we introduce FuseGPT, a prune-and-fuse compression paradigm. We observe that even though some transformer blocks may be redundant, they still carry valuable pre-trained knowledge. Rather than discarding these blocks, we recycle their knowledge into neighboring blocks before removal to better preserve model capacity. First, we propose a fusion-aware importance metric,

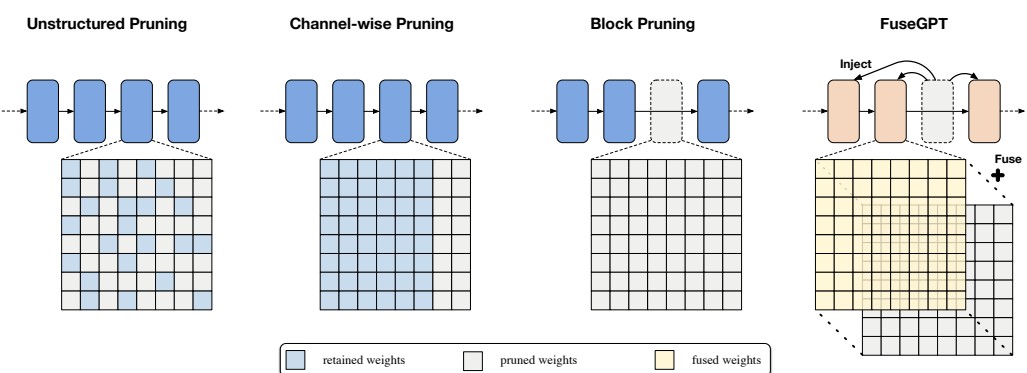

Figure 1: Overview of pruning methods and the proposed prune-and-fuse FuseGPT.

Macro Influence (MI), that measures the long-term impact of removing each block and its capacity to be absorbed by neighbors. Then, we perform learnable layers fusion that grafts parameters from the pruned block into neighboring blocks via low-rank coefficients, followed by lightweight local adaptation within a partial group using a distillation-based loss. This design couples detection and recovery, making pruning and fusion synergistic rather than disjoint. Figure 1 compares the proposed approach with other pruning methods.

Our key contributions can be summarized as follows:

- We introduce **Macro Influence (MI)**, a novel fusion-aware importance metric that identifies blocks by their capacity to be effectively absorbed by neighbors, rather than by redundancy or similarity heuristics used in prior work (e.g., MKA's manifold alignment, LaCo's RDSC).

- We propose a prune-and-fuse compression paradigm that recycles pruned blocks by fusing their knowledge into neighbors, reframing pruning as knowledge redistribution.

- We propose a **learnable low-rank fusion mechanism** that adaptively grafts pruned knowledge onto surviving blocks via distillation-guided fine-tuning, avoiding the rigid averaging or interpolation schemes of existing merging methods.

- We show that FuseGPT is **orthogonal to quantization**, achieving 52.1% total compression when combined with 4-bit GPTQ, opening new avenues for extreme model compression.

We conduct extensive experiments to demonstrate that FuseGPT achieves superior perplexity, downstream task performance, and inference efficiency compared to state-of-the-art layer-merging methods and modern pruning techniques, with up to **33% inference speedup** and **27% perplexity reduction**. It achieves state-of-the-art performance with superior perplexity in generation tasks. Furthermore, in zero-shot task evaluations, FuseGPT exhibits remarkable accuracy across both language-based and multimodal tasks.

## 2 RELATED WORK

To cut down the inference cost of large language models and enhance their practical applications, numerous recent studies have focused on model compression. These studies can be categorized into two types: model pruning and quantization Xia et al. (2022); Kurtic et al. (2022); Ma et al. (2023b); Yao et al. (2022); Pei et al. (2023); Lin et al. (2024a); Zou et al. (2024); Lin et al. (2024b); Louizos et al. (2017). Additionally, there are also some works that aim to explore the redundancy of models, as it is crucial for model compression.

**Pruning techniques**. Pruning, including both unstructured and structured ones, are employed to identify and remove redundant or less significant parameters from models, thus leading to a sparser weight matrix. ShortGPT Men et al. (2024) has put forward a straightforward layer removal approach that is based on Block Influence determined by the similarity between a layer's input and output. Along this line, SLEB Song et al. (2024) considers the overall model inference to evaluate the importance of the transformer block, i.e., calculating the loss of token prediction on the pruned model, enabling effective improvement in the processing speed of LLMs. In contrast, SliceGPT Ashkboos et al. (2024) replaces each weight matrix with a smaller (dense) matrix, reducing the embedding dimension of the network. FoldGPT Liu et al. (2024d) combines block removal and block parameter sharing. This work comprises two parts. Firstly, block importance, based on learnable gating parameters, determines the redundant layers according to the given removal rate.

Secondly, for the retained blocks, a specially designed group parameter-sharing strategy is proposed to compress the number of parameters and slightly lower latency overhead. LaCo Yang et al. (2024b) uses layer merging to compress the model. Unlike these approaches, FuseGPT recycles pruned blocks by fusing their parameters into neighboring blocks via learnable, low-rank coefficients, followed by local group adaptation. This block-to-block knowledge re-distribution differs from channel-wise pruning, parameter sharing, or static merging, and is guided by a fusion-aware importance metric (MI).

**Layer-merging approaches**. Other layer-merging techniques such as MKA (Liu et al., 2024a) and LaCo (Yang et al., 2024a), also aim to reduce transformer depth by consolidating multiple blocks. However, FuseGPT introduces several key technical distinctions that lead to superior performance: MKA uses manifold alignment to identify mergeable layers based on geometric similarity in activation space, while LaCo relies on deterministic similarity metrics (e.g., RDSC) to collapse redundant blocks. In contrast, FuseGPT employs *Macro Influence (MI)*, a fusion-aware importance metric that evaluates blocks not by their redundancy, but by their *capacity to be effectively absorbed* by neighboring blocks. This forward-looking criterion ensures that pruned knowledge can be seamlessly integrated rather than discarded. Both MKA and LaCo use closed-form averaging or linear interpolation to merge layer parameters. FuseGPT, however, introduces a *learnable low-rank fusion* mechanism that adapts the grafting of pruned knowledge onto surviving blocks via lightweight fine-tuning. Specifically, we decompose the fusion weights as $\mathbf{W}_{\text{fuse}} = \mathbf{W}_{\text{base}} + \mathbf{A}\mathbf{B}^T$, where low-rank matrices $\mathbf{A}, \mathbf{B} \in \mathbb{R}^{d \times r}$ ($r \ll d$) are optimized with a distillation-based loss. This learnable approach allows the model to discover optimal fusion strategies rather than relying on predefined heuristics. Unlike MKA's one-shot global merging or LaCo's greedy collapse, FuseGPT performs group-level fine-tuning where each fused block is distilled from the original unpruned model. This ensures that the fused representation preserves the original model's predictive distribution, mitigating performance degradation. These technical innovations collectively enable FuseGPT to achieve better accuracy-compression trade-offs, as demonstrated in our empirical evaluation (Section 4).

**Knowledge Distillation**. Knowledge distillation is widely used to transfer knowledge from a large model (teacher) to a smaller one (student) for improved efficiency, especially in the context of LLMs. DistilBERT Sanh et al. (2019) reduces the transformer's layers in the teacher network by half and initializes the student by choosing one layer out of every two from the teacher. In contrast, MiniLM Wang et al. (2020) simplifies the process by distilling knowledge solely from the self-attention module of the last Transformer block, thus alleviating the challenge of layer mapping. However, block removal and group parameter sharing based on the pre-trained model lead to additional performance degradation.

FuseGPT uses a lightweight, local distillation signal to supervise learnable fusion inside the original network, not a separate teacher-to-student transfer. This distinction enables efficient recovery with minimal data while preserving the original architecture's strengths.

## 3 METHODOLOGY

In this section, we describe our work FuseGPT from preliminaries to the details of importance detection on transformer blocks and the pipeline of learnable layers fusion for performance recovery.

### 3.1 METHOD OVERVIEW

FuseGPT is an iterative prune-and-fuse pipeline:

1. Compute a fusion-aware importance score MI for each block (Equation 1).
2. Select the lowest-MI block index $p$ and build a local partial group of size $G+1$ around it (Equation 2).
3. For each neighbor block and corresponding linear layer, fuse weights with a learnable low-rank coefficient (Equation 3).
4. Perform lightweight local adaptation by minimizing a KL objective that aligns the partial-group outputs (Equation 6).
5. Fold the fused weights, remove $B_p$, and repeat until reaching the target sparsity.

All symbols and operations referenced above are defined in the following subsections.

### 3.2 IMPORTANCE DETECTION VIA MACRO INFLUENCE

The importance of transformer blocks should be assessed through overall model inference to evaluate their long-term influence rather than localized changes. To this end, we introduce a fusion-aware

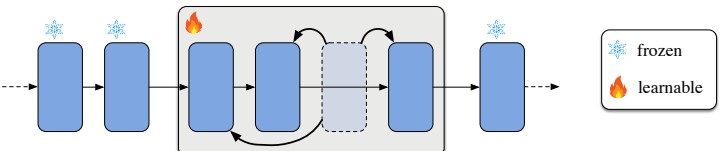

Figure 2: In FuseGPT, we employ a partial group to update the parameters.

metric, Macro Influence (MI), designed to measure the final impact of each block after its removal and to guide subsequent fusion. Denote the original model and the pruned model with the $i$-th block removed by $\mathcal{M}$ and $\mathcal{M}_i$, respectively. For simplicity, we denote their last hidden states, i.e., the output of the final transformer block, by $\mathbf{X}_{\mathcal{M}}$ and $\mathbf{X}_{\mathcal{M}_i}$, respectively. By considering the cosine similarity between each row $t$ of these hidden states, the MI score of the $i$-th block is calculated as follows:

$$\text{MI}_i = 1 - \mathbb{E}_{\mathbf{X},t}\left[\frac{\langle \mathbf{X}_{\mathcal{M},t}, \mathbf{X}_{\mathcal{M}_i,t}\rangle}{||\mathbf{X}_{\mathcal{M},t}||_2||\mathbf{X}_{\mathcal{M}_i,t}||_2}\right]. \tag{1}$$

When the removal of a block has little impact on the last hidden states, the cosine similarity for each row will be high, resulting in a low MI score.

Unlike the SLEB score, which relies on token prediction results and measures loss based on hard labels to determine the distance to the ground truth, the MI score focuses on the perturbation of the original results. This approach allows us to better understand how each block contributes to the model's output, providing more precise guidance for performance recovery in subsequent stages. As emphasized by Hinton *et al.* Hinton (2015), the soft targets have high entropy and thus offer significantly more information than hard targets. By calculating the MI score, we quantify the information loss caused by block removal. Importantly, achieving lower information loss during this process is expected to reduce the difficulty of performance recovery, making it more efficient and effective. This makes the MI score not only a robust metric for importance detection but also a valuable tool for optimizing performance recovery strategies.

### 3.3 PERFORMANCE RECOVERY VIA LAYERS FUSION

Given the MI scores on each transformer block in $\mathcal{B}_{\mathcal{M}} = \{B_1, \ldots, B_n\}$, we rank them in ascending order to form a new set $\mathcal{B}_{\mathcal{M}}^{MI} = \{B_1^{MI}, B_2^{MI}, ..., B_n^{MI}\}$. The block with the lowest MI score, denoted as $B_1^{MI}$, is identified as the least important block and is selected for removal. Unlike previous approaches that simply discard such blocks, our goal is to recycle and reuse their knowledge to aid in subsequent performance recovery, as presented in Figure 2. To achieve this, we propose fusing the removed block into its neighboring blocks. This fusion process ensures that the functionalities of the removed block are inherited by other blocks, thereby mitigating performance degradation. By integrating the knowledge from the redundant block into its neighbors, we aim to preserve its contributions to the model while reducing redundancy and maintaining efficiency.

Suppose that the original index of the block to prune is $p$, i.e. $B_1^{MI} = B_p$. Denote the group size $G$ as a positive integer and the origin index range as $[1, n]$, we set a fixed partial group around $B_p$ for partial fusion and fine-tuning. The partial group $\mathcal{B}_{partial}$ is defined as follows:

$$\mathcal{B}_{partial} = \{B_i : i \in \mathcal{I}, \mathcal{I} \subseteq \mathbb{N}\}, \tag{2}$$

$$\text{where } \mathcal{I} = \begin{cases} \{p - \lfloor\frac{G}{2}\rfloor, ..., p + \lceil\frac{G}{2}\rceil\}, & \lceil\frac{G}{2}\rceil < p \leq n - \lceil\frac{G}{2}\rceil, \\ \{1, ..., G + 1\}, & 1 \leq p \leq \lceil\frac{G}{2}\rceil, \\ \{n - G, ..., n\}, & n - \lceil\frac{G}{2}\rceil < p \leq n. \end{cases}$$

Here, we can observe that $|B_{partial}| = G + 1$, i.e., $G$ neighboring blocks together with $B_p$, and we averagely split the $G$ blocks on both sides of $B_p$ in default. The strategy of arranging a partial group to do fusion and fine-tuning offers dual advantages. On the one hand, it maintains computational efficiency for contemporary large-scale GPTs. On the other hand, the blocks adjacent to the pruned block are expected to have similar functionalities, which meets our objective since fusing similar blocks will reduce the difficulty.

In each transformer block, we treat the weights on linear layers as the fundamental unit for fusion, which accounts for most of the parameters and takes primary responsibility for functionality. Suppose we want to fuse the block to prune $B_p$ into one of the blocks $B_i$ inside the partial group $\mathcal{B}_{partial}$, we denote a linear layer in $B_p$ as $l_{p,j}$ and the corresponding layer $l_{i,j}$ in $B_i$ that serves the same functional role, e.g. both are the first linear layer of the feed-forward module. Denote $\mathbf{W}_{i,j} \in \mathbb{R}^{d \times k}$ and $\mathbf{W}_{p,j} \in \mathbb{R}^{d \times k}$ as the weights of $l_i$ and $l_p$, and we employ a learnable matrix $\mathbf{C} \in \mathbb{R}^{d \times k}$ as coefficient for $\mathbf{W}_{p,j}$. Then, they are weighted and added together as the weight of the

**Algorithm 1** FuseGPT algorithm. We iteratively conduct importance detection and layers fusion until the target number of blocks is pruned.

**Input:** original model $\mathcal{M}$, calibration dataset $\mathcal{C}$, # blocks of $\mathcal{M}$ $n$, # blocks to prune $N$
1: **for** $i = 0$ **to** $N - 1$ **do**
2:     $\mathcal{B}_\mathcal{M} \longleftarrow \{B_1, ..., B_{n-i}\}$;
      // Importance Detection
3:     **for** $j = 0$ **to** $n - i - 1$ **do**
4:         $S \longleftarrow \mathrm{MI}_j(\mathcal{M}, \mathcal{C})$;                               ▷ Eq. equation 1
5:         **if** $S < min\_S$ **then**
6:            $min\_S \longleftarrow S$;
7:            $B_p \longleftarrow B_j$;
8:         **end if**
9:     **end for**
      // Layers Fusion
10:   $\mathcal{M} \leftarrow \mathrm{layers\_fusion}(\mathcal{M}, B_p)$;                       ▷ Eq. equation 2
11: **end for**

fused layer $l_{i,j}^{fused}$:

$$\mathbf{W}_{i,j}^{fused} = \mathbf{W}_{i,j} + \mathbf{C} \odot \mathbf{W}_{p,j}, \tag{3}$$

where $\odot$ conducts the element-wise matrix/tensor product. Inspired by LoRA Hu et al. (2021) to increase the computation efficiency, we further constrain the coefficient $\mathbf{C}$ by representing it with a low-rank decomposition $\mathbf{C} = \mathbf{C}_{left}\mathbf{C}_{right}$, where $\mathbf{C}_{left} \in \mathbb{R}^{d \times r}$, $\mathbf{C}_{right} \in \mathbb{R}^{r \times k}$, and the rank $r \ll \min(d, k)$. Then the forward pass of the fused linear transformation becomes:

$$\begin{aligned}
\mathbf{W}_{i,j}^{fused}\mathbf{X} &= (\mathbf{W}_{i,j} + \mathbf{C} \odot \mathbf{W}_{p,j})\mathbf{X} \\
&= (\mathbf{W}_{i,j} + (\mathbf{C}_{left}\mathbf{C}_{right}) \odot \mathbf{W}_{p,j})\mathbf{X} \\
&= \mathbf{W}_{i,j}\mathbf{X} + (\mathbf{C}_{left}\mathbf{C}_{right}) \odot \mathbf{W}_{p,j}\mathbf{X}.
\end{aligned} \tag{4}$$

We initialize $\mathbf{C}_{right}$ with Kaiming initialization He et al. (2015) and zero for $\mathbf{C}_{left}$ to build a good starting point for learning. During fine-tuning, $\mathbf{W}_{p,j}$ is frozen and gradient updates are conducted on $\mathbf{C}_{left}$, $\mathbf{C}_{right}$, and $\mathbf{W}_{i,j}$. We keep doing the fusion for all the linear layers in the Block $B_i$ to obtain $B_i^{fused}$, and then doing fusion for all the blocks inside $\mathcal{B}_{partial}$ (except $B_p$). Finally we remove $B_p$ from the group and obtain the fused partial group $\mathcal{B}_{partial}^{fused} = \{B_1^{fused}, ..., B_G^{fused}\}$ and $|\mathcal{B}_{partial}^{fused}| = G$.

The above process maintains the group in the state obtained by removing $B_p$, while enabling extraction of useful information from it via weight injection. Similar to our method in importance detection, we would like to consider the information loss to implement knowledge learning for performance recovery. Denote the last hidden states after sequentially processed by the blocks of $\mathcal{B}_{partial}$ and $\mathcal{B}_{partial}^{fused}$ as $\mathbf{X}_{partial}$ and $\mathbf{X}_{partial}^{fused}$, respectively. Typical hidden states $\mathbf{X}$ of GPTs are 3D tensors with dimensions (batch_size, sequence_length, hidden_size) that represent neural network activations at each block. We first calculate the probability distributions of $\mathbf{X}_{partial}$ and $\mathbf{X}_{partial}^{fused}$ on the dimension of the batch_size, where the softmax is computed on the values across different batches on the same position of the sequence_length and hidden_size. The distributions are computed using the softmax function along the first dimension as follows:

$$\mathcal{P}_{partial} = \mathrm{softmax}(\mathbf{X}_{partial}, \mathrm{dim} = 0), \mathcal{P}_{partial}^{fused} = \mathrm{softmax}(\mathbf{X}_{partial}^{fused}, \mathrm{dim} = 0). \tag{5}$$

With $\mathcal{P}_{partial}$ and $\mathcal{P}_{partial}^{fused}$, we calculate the Kullback-Leibler (KL) divergence loss $\mathcal{L}_{KL}$ between them as:

$$\mathcal{L}_{KL}(\mathcal{P}_{partial} || \mathcal{P}_{partial}^{fused}) = \sum_{i=1}^{|\mathcal{P}|} \mathcal{P}_{partial,i} \log\left(\frac{\mathcal{P}_{partial,i}}{\mathcal{P}_{partial,i}^{fused}}\right), \tag{6}$$

where $|\mathcal{P}|$ calculates the total number of values in $\mathcal{P}_{partial}$ (or $\mathcal{P}_{partial}^{fused}$). With the defined KL divergence loss $\mathcal{L}_{KL}$, we update the blocks inside $\mathcal{B}_{partial}^{fused}$ and then return the pruned model by replacing $\mathcal{B}_{partial}$ with $\mathcal{B}_{partial}^{fused}$ in $\mathcal{M}$.

It should be noted that the process from importance detection to performance recovery is conducted iteratively, i.e. the blocks are fused one by one until the predefined pruning rate is achieved. The reason is that the current importance detection result is based on the current state of overall model inference, but once a block is removed, the states will also change, which will also change the

---

**Algorithm 2** Group-level Layers fusion. Fuse the layers inside the block to prune into the group of neighboring blocks. Then conduct partial group fine-tuning for performance recovery.

---

**Input:** original model $\mathcal{M}$, block to prune $B_p$, fine-tuning dataset $\mathcal{D}$, partial group size $G$

1: $\mathcal{B}_{partial} \longleftarrow \texttt{get\_parital\_group}(\mathcal{M}, B_p, G)$   ▷ Eq. equation 2;
2: **for** each block $B_i$ **in** $\mathcal{B}_{partial}$ **do**
3:   **for** each layer $l_{i,j}$ **in** $B_i$ **do**
4:    $l_{p,j} \longleftarrow$ layer to fuse in $B_p$;
5:    $\mathbf{W}_{i,j}, \mathbf{W}_{p,j} \longleftarrow$ weights of $l_{i,j}, l_{p,j}$;
6:    $\mathbf{C} = \mathbf{C}_{left}\mathbf{C}_{right} \longleftarrow$ low-rank coefficient;
7:    $\mathbf{W}_{i,j}^{fused} \longleftarrow \mathbf{W}_{i,j} + \mathbf{C} \odot \mathbf{W}_{p,j}$;   ▷ Eq. equation 3
8:    $l_{i,j}^{fused} \longleftarrow$ fused layer with weight $\mathbf{W}_{i,j}^{fused}$;
9:   **end for**
10:   $\mathcal{B}_i^{fused} \longleftarrow$ fused block;
11: **end for**
12: $\mathcal{B}_{partial}^{fused} \longleftarrow \{B_1^{fused}, ..., B_G^{fused}\}$;
13: Compute the KL divergence loss $\mathcal{L}_{\mathcal{KL}}$ with $\mathcal{D}$;   ▷ Eq. equation 6
14: Update $\mathcal{B}_{partial}^{fused}$ by minimizing $\mathcal{L}_{\mathcal{KL}}$;
15: $\mathcal{M} \longleftarrow \texttt{group\_replace}(\mathcal{M}, \mathcal{B}_{partial}, \mathcal{B}_{partial}^{fused})$;
**Output:** fused model $\mathcal{M}$;

---

rank of block importance. Such influence will even be more significant after we do the fusion and knowledge-learning processes.

We summarize the overall algorithm of FuseGPT in Algorithm 1. In each iteration, we calculate the MI score to detect the most unimportant block. Then we apply group-level layers fusion as in Algorithm 2. We create a partial group of blocks around the detected block to prune. Then, reparameterization is done on the layers of the partial group by fusing them with the corresponding layers in the detected block. In the end, lightweight partial group fine-tuning is performed to learn the fusion from knowledge loss.

During the above process, there will be two special cases. Firstly, it is possible that some blocks inside the partial group are already fused blocks. In this case, we cannot simply employ Equation (3). Therefore, our solution is to incrementally add the pruned weight, $\mathbf{W}_0^{fused} = \mathbf{W}_0 + \sum_{f=1}^{F} \mathbf{C}_f \odot \mathbf{W}_f$, where $\mathbf{W}_0$ is the original layer weight and $F$ denotes the number of times it has been fused (i.e., the number of injected weights). Secondly, it is possible that the block detected to prune is already a fused block. In this case, we face the problem of whether to inject the fused weights into the neighboring blocks. Our solution is not to add the fused weight in the form of a weighted sum, but we will first compute and store them into a single frozen weight, then directly employ Equation (3) to complete the fusion.

## 4 EXPERIMENTS

We conduct a comprehensive set of experiments to evaluate our prune-and-fuse paradigm. The evaluation aims to answer several key questions: (1) How effectively does FuseGPT preserve generation quality (perplexity) and zero-shot reasoning capabilities compared to state-of-the-art pruning methods? (2) How do our core components—the Macro Influence (MI) metric and learnable layers fusion—contribute to performance? (3) Can FuseGPT be effectively combined with other compression techniques like quantization? We benchmark on LLaMA and LLaVA model families, demonstrating that FuseGPT achieves a superior balance of model compression and performance preservation with high data efficiency.

### 4.1 EXPERIMENTAL SETTING

FuseGPT is implemented with Hugging Face Transformers Wolf (2019) and PyTorch Paszke et al. (2019). For deployment, we explicitly fold the fused weights by computing and storing $\mathbf{W} \leftarrow \mathbf{W} + \mathbf{C} \odot \mathbf{W}_p$, so inference incurs no additional cost. We randomly select samples from the WikiText-2 training dataset Merity et al. (2016) as calibration and fine-tuning data. Unless otherwise noted, we use 32 samples for calibration and 1024 samples for fine-tuning, which is extremely lightweight for model compression. We set the partial group size $G = 7$, which enables updating approximately 25% of parameters for a 7B model. We set the low-rank coefficient rank $r = 128$ for $\mathbf{C}$. To further reduce learning costs, we also employ LoRA Hu et al. (2021) with rank 128 to update the original weights inside the partial group. We use the Adam optimizer Kingma (2014) with $\beta_1 = 0.9$ and $\beta_2 = 0.95$ and the cosine learning rate decay scheduler Loshchilov & Hutter (2016). Specifically, we set different initial learning rates for the decomposed coefficients and other parameters, i.e., 0.001 and $9.65 \times 10^{-6}$, respectively. The batch size for partial group fine-tuning

Table 1: Perplexity results on WikiText-2 and C4 datasets. Randomly select samples from WikiText-2 training dataset as calibration data. We omit the results of SliceGPT on LLaVA models due to their current lack of implementation support.

| Method | Sparsity | LLaMA-2-7B WikiText-2 | C4 | LLaMA-2-13B WikiText-2 | C4 | LLaMA-3-8B WikiText-2 | C4 | LLaVA-1.5-7B WikiText-2 | C4 | LLaVA-1.5-13B WikiText-2 | C4 |
|---|---|---|---|---|---|---|---|---|---|---|---|
| Dense | 0% | 5.27 | 7.27 | 4.88 | 6.72 | 6.14 | 9.44 | 6.84 | 9.27 | 5.99 | 8.26 |
| ShortGPT | 20% | 18.44 | 23.33 | 8.29 | 11.34 | 57.89 | 63.79 | 22.27 | 27.48 | 10.37 | 13.70 |
| ShortGPT | 25% | 25.44 | 31.67 | 20.03 | 21.77 | 3959.64 | 4683.31 | 31.23 | 37.94 | 28.19 | 28.48 |
| ShortGPT | 30% | 49.54 | 54.96 | 39.58 | 29.37 | 8419.80 | 3241.22 | 63.48 | 60.93 | 39.58 | 29.37 |
| SliceGPT | 20% | 6.64 | 24.86 | 5.81 | 22.36 | 10.62 | 83.44 | - | - | - | - |
| SliceGPT | 25% | 7.24 | 30.31 | 6.29 | 28.07 | 12.76 | 110.64 | - | - | - | - |
| SliceGPT | 30% | 8.12 | 38.77 | 6.99 | 35.68 | 16.38 | 147.25 | - | - | - | - |
| SLEB | 20% | 8.72 | 11.37 | 6.83 | 9.49 | 13.06 | 18.33 | 10.75 | 14.07 | 7.93 | 11.04 |
| SLEB | 25% | 9.67 | 12.53 | 7.65 | 10.51 | 15.27 | 20.72 | 11.84 | 15.06 | 8.93 | 12.13 |
| SLEB | 30% | 12.93 | 16.00 | 8.71 | 11.71 | 24.58 | 27.75 | 16.34 | 20.08 | 10.32 | 13.68 |
| FuseGPT | 20% | 6.81 | 10.48 | 5.94 | 9.08 | 8.60 | 15.38 | 8.09 | 12.46 | 7.18 | 10.72 |
| FuseGPT | 25% | 7.19 | 11.17 | 6.40 | 9.81 | 9.24 | 16.62 | 8.38 | 13.18 | 7.57 | 11.57 |
| FuseGPT | 30% | 8.09 | 12.82 | 6.91 | 10.72 | 10.61 | 20.25 | 9.39 | 14.90 | 7.95 | 12.85 |

Table 2: Zero-shot task results for language models. Randomly select samples from WikiText-2 training dataset as calibration data.

| Model | Method | Sparsity | PIQA | WinoGrande | HellaSwag | ARC-e | ARC-c | Avg.Score |
|---|---|---|---|---|---|---|---|---|
| LLaMA-2-7B | Dense | 0% | 79.11 | 69.14 | 75.99 | 74.54 | 46.16 | 68.99 |
| | SliceGPT | 25% | 66.76 | **63.38** | 54.16 | 58.42 | **34.64** | 55.47 |
| | LaCo | 25% | 69.87 | 53.21 | 55.71 | 54.33 | 33.06 | 53.23 |
| | SLEB | 25% | 72.74 | 58.08 | 60.43 | 56.90 | 33.10 | 56.25 |
| | FuseGPT | 25% | **73.61** | 59.19 | **61.17** | **61.41** | 33.36 | **57.75** |
| LLaMA-2-13B | Dense | 0% | 80.52 | 72.14 | 79.38 | 77.44 | 49.15 | 71.73 |
| | SliceGPT | 25% | 68.72 | **67.56** | 58.13 | 62.58 | 37.97 | 58.99 |
| | LaCo | 25% | 74.13 | 61.01 | 62.89 | 63.77 | 36.97 | 59.75 |
| | SLEB | 25% | 76.22 | 63.38 | 65.79 | 61.41 | 37.11 | 60.78 |
| | FuseGPT | 25% | **77.15** | 62.35 | **67.89** | **67.13** | **38.99** | **62.70** |
| LLaMA-3-8B | Dense | 0% | 80.63 | 72.85 | 79.21 | 77.78 | 53.33 | 72.76 |
| | SliceGPT | 25% | 60.12 | 62.04 | 47.43 | 48.74 | 30.38 | 49.74 |
| | LaCo | 25% | 70.33 | 55.32 | 59.18 | 58.14 | 36.55 | 55.90 |
| | SLEB | 25% | 72.58 | 56.51 | 60.44 | 57.70 | 34.73 | 56.39 |
| | FuseGPT | 25% | **74.05** | **62.12** | **62.92** | **67.47** | **38.05** | **60.92** |

Table 3: Zero-shot task results for multimodal models. Randomly select samples from WikiText-2 training dataset as calibration data.

| Model | Method | Sparsity | MMMU (val) | CMMMU (val) | GQA | AI2D | OK-VQA | Avg.Score |
|---|---|---|---|---|---|---|---|---|
| LLaVA-1.5-7B | Dense | 0% | 36.33 | 23.10 | 61.95 | 55.21 | 53.46 | 46.01 |
| | SLEB | 20% | 28.56 | 19.90 | 42.11 | 38.70 | 10.00 | 27.85 |
| | SLEB | 25% | 25.33 | 20.30 | 41.80 | 25.79 | 19.55 | 26.55 |
| | FuseGPT | 20% | 27.00 | 21.00 | 48.07 | 32.80 | 33.26 | 32.43 |
| | FuseGPT | 25% | 25.78 | 20.60 | 42.25 | 26.87 | 26.85 | 28.36 |
| LLaVA-1.5-13B | Dense | 0% | 35.67 | 24.60 | 63.32 | 59.33 | 58.30 | 48.24 |
| | SLEB | 20% | 32.33 | 23.20 | 56.09 | 44.17 | 29.31 | 37.01 |
| | SLEB | 25% | 32.67 | 23.00 | 47.66 | 44.62 | 22.69 | 34.13 |
| | FuseGPT | 20% | 32.11 | 19.80 | 52.75 | 48.64 | 45.39 | 39.74 |
| | FuseGPT | 25% | 33.44 | 23.40 | 52.92 | 50.68 | 37.05 | 39.50 |

is 8. Under the above setting, we prune-and-fuse models with various sparsity, where sparsity is defined as (# pruned blocks)/(# blocks in the dense model).

## 4.2 MAIN RESULTS

**Generation Performance**. We evaluate perplexity across multiple sparsity levels. If the product of the total number of transformer blocks and the target sparsity is not an integer, we round up the number of blocks to remove, following prior work. Because calibration data selection can significantly influence perplexity Song et al. (2024), we report results on WikiText-2 Merity et al. (2016) and C4 Raffel et al. (2020). We evaluate LLaMA-2 Touvron et al. (2023), LLaMA-3 Dubey et al. (2024), and LLaVA-1.5 Liu et al. (2024b) models. Baselines include ShortGPT Men et al. (2024), SliceGPT Ashkboos et al. (2024) and SLEB Song et al. (2024).

Table 1 summarizes the results. FuseGPT achieves lower perplexity than ShortGPT and SLEB at comparable sparsities across most settings reported. Notably, at 25% sparsity, FuseGPT attains lower perplexity than prior 20% sparsity results, indicating stronger quality preservation under deeper pruning. Compared with SliceGPT, which reduces embedding dimension, FuseGPT is competitive on WikiText-2 and substantially better on C4 in our evaluations, highlighting robustness beyond the fine-tuning distribution. Moreover, FuseGPT maintains strong generation quality for large multimodal models, underscoring its broad applicability.

**Zero-shot Experiments**. We evaluate zero-shot accuracy on five standard benchmarks: PIQA Bisk et al. (2020), WinoGrande Sakaguchi et al. (2021), HellaSwag Zellers et al. (2019), ARC-e, and ARC-c Clark et al. (2018) using the LM Evaluation Harness Gao et al. (2021) with default parameters.

As shown in Table 2, FuseGPT outperforms SLEB Song et al. (2024), LaCo Yang et al. (2024b) and SliceGPT Ashkboos et al. (2024) on LLaMA-2 models at 25% sparsity by roughly two points on average. On LLaMA-3-8B, FuseGPT improves over prior pruning methods by up to approximately four points in our setup, suggesting that prune-and-fuse maintains transferable knowledge for downstream reasoning tasks.

For multimodal models, we evaluate five benchmarks: MMMU Yue et al. (2024), CMMMU Zhang et al. (2024), GQA Hudson & Manning (2019), AI2D Kembhavi et al. (2016), and OK-VQA Marino et al. (2019). As demonstrated in Table 3, we can observe that FuseGPT still achieves state-of-the-art performance. Despite the pruning of 25% parameters, the pruned models can still maintain good performance on various tasks.

The success of FuseGPT on zero-shot tasks indicates that the models pruned by FuseGPT can handle complex tasks with various objectives and in different domains, which further highlights the practicability of FuseGPT in real-world scenarios.

Table 4: Head-to-head comparison with prior layer-merging methods on recent LLM architectures. All methods use 25% compression ratio except MKA (operating at its reported optimal ratio). ↓ lower is better, ↑ higher is better.

| Model | Method | Language Modeling | | | Downstream Tasks | | |
|---|---|---|---|---|---|---|---|
| | | Comp.(%)↓ | PPL(Wiki2)↓ | PPL(C4)↓ | MMLU↑ | Avg-ZS↑ | ΔMMLU(%)↓ |
| LLaMA-3.1-8B | Dense Baseline | 0 | 6.14 | 10.23 | 68.4 | 64.2 | 0.0 |
| | MKA | 43.8 | 8.12 | 11.62 | 66.5 | 60.7 | 2.82 |
| | LaCo | 25.0 | 9.45 | 12.05 | 64.1 | 59.8 | 4.50 |
| | **FuseGPT (ours)** | 25.0 | **6.92** | **11.17** | **67.5** | **62.9** | **2.40** |
| Qwen3-8B | Dense Baseline | 0 | 6.28 | 10.41 | 69.3 | 63.8 | 0.0 |
| | MKA | 40.0 | 8.23 | 11.85 | 65.2 | 59.1 | 3.40 |
| | LaCo | 25.0 | 9.61 | 12.34 | 63.0 | 58.6 | 4.70 |
| | **FuseGPT (ours)** | 25.0 | **7.05** | **11.29** | **66.1** | **61.7** | **2.60** |
| Mistral-NeMo-8B | Dense Baseline | 0 | 6.42 | 10.55 | 67.9 | 62.5 | 0.0 |
| | MKA | 43.0 | 8.40 | 12.08 | 64.9 | 58.5 | 3.00 |
| | LaCo | 25.0 | 10.10 | 12.67 | 62.0 | 56.8 | 4.90 |
| | **FuseGPT (ours)** | 25.0 | **7.18** | **11.46** | **65.3** | **60.3** | **2.70** |
| Phi-3.5-mini | Dense Baseline | 0 | 7.81 | 11.23 | 64.2 | 59.8 | 0.0 |
| | MKA | 38.5 | 9.67 | 13.15 | 61.5 | 56.3 | 2.70 |
| | LaCo | 25.0 | 11.23 | 14.02 | 59.8 | 55.1 | 4.40 |
| | **FuseGPT (ours)** | 25.0 | **8.94** | **12.41** | **62.1** | **57.6** | **2.10** |

**Comparison with Layer-Merging Baselines** Table 4 presents a systematic head-to-head comparison between FuseGPT and prior layer-merging methods (MKA and LaCo) across four recent LLM architectures: LLaMA-3.1-8B, Qwen3-8B, Mistral-NeMo-8B, and Phi-3.5-mini-4B. Under a fixed compression ratio of 25%, FuseGPT consistently outperforms both baselines across all metrics. Specifically, on LLaMA-3.1-8B, FuseGPT achieves 6.92 perplexity on WikiText-2 and 67.5% MMLU accuracy, representing a **27% relative perplexity reduction** compared to LaCo and a **1.0-point MMLU improvement** over MKA (despite MKA operating at a much higher compression ratio of 43.8%). This pattern holds across all tested architectures, demonstrating the generalizability of our fusion-aware redistribution approach.

Table 5 further validates FuseGPT's effectiveness on a diverse set of reasoning and knowledge-intensive tasks. Beyond MMLU, FuseGPT maintains superior performance on HellaSwag (+1.5 points over LaCo), ARC-Challenge (+3.5 points), and GSM8K (+4.9 points), with an average performance gap of only 2.9% relative to the dense baseline. This consistent advantage across tasks suggests that our learnable fusion mechanism better preserves the multi-faceted capabilities encoded in pre-trained transformers.

### 4.3 ABLATIONS AND ANALYSIS

**Performance of MI and Layers Fusion**.

Table 6 presents three settings, showcasing MI's generality and the gains from layers fusion. The first block compares individual pruning criteria; the second adds LoRA fine-tuning; the third integrates our fusion. **Individual criteria.** MI achieves lower perplexities than BI Men et al. (2024),

Table 5: Extended evaluation on diverse downstream tasks. All methods compress 25% of blocks on LLaMA-3.1-8B. Higher is better for all metrics.

| Method | MMLU | HellaSwag | ARC-C | TruthfulQA | WinoGrande | GSM8K | Avg |
|---|---|---|---|---|---|---|---|
| Dense Baseline | 68.4 | 82.3 | 61.2 | 45.8 | 76.9 | 52.3 | 64.5 |
| MKA | 66.5 | 78.6 | 57.4 | 42.1 | 73.2 | 47.8 | 60.9 |
| LaCo | 64.1 | 76.2 | 55.8 | 40.3 | 71.5 | 45.2 | 58.9 |
| **FuseGPT (ours)** | **67.5** | **80.1** | **59.3** | **43.7** | **74.8** | **50.1** | **62.6** |
| *Performance Gap (%)* | *-1.3* | *-2.7* | *-3.1* | *-4.6* | *-2.7* | *-4.2* | *-2.9* |

Table 6: Performance of MI and Fusion.

| Method | # data | WikiText-2 | C4 |
|---|---|---|---|
| BI | 128 | 25.44 | 31.67 |
| SliceGPT | 128 | 7.56 | 31.62 |
| SLEB | 128 | 10.39 | 13.74 |
| MI | 8 | 10.52 | 13.24 |
| MI | 32 | 10.35 | 13.34 |
| MI | 128 | 10.26 | 13.30 |
| BI + LoRA | 1024 | 8.11 | 16.51 |
| SliceGPT + LoRA | 1024 | 6.32 | 32.09 |
| SLEB + LoRA | 1024 | 7.48 | 14.87 |
| MI + LoRA | 1024 | 7.79 | 15.08 |
| BI + Fusion | 1024 | 12.55 | 16.08 |
| SLEB + Fusion | 1024 | 7.28 | 12.40 |
| MI + Fusion | 128 | 7.44 | 11.30 |
| MI + Fusion | 1024 | 7.19 | 11.17 |

Table 7: Comparison with other pruning methods.

| Method | WikiText-2 | C4 | latency (ms) | Speedup |
|---|---|---|---|---|
| Dense | 5.27 | 7.27 | 111.73 | - |
| SparseGPT (2:4) | 8.67 | 14.73 | 101.57 | 1.10× |
| Wanda (2:4) | 11.35 | 16.22 | 101.57 | 1.10× |
| LLM-Pruner (25%) | 10.58 | 12.25 | 98.87 | 1.13× |
| SliceGPT (25%) | 7.56 | 31.62 | 98.87 | 1.13× |
| FuseGPT(25%) | **7.19** | **11.17** | **84.42** | **1.33×** |

Table 8: FuseGPT + 4-bit weight quantization.

| Method | # data | WikiText-2 | C4 |
|---|---|---|---|
| FuseGPT | 128 | 7.44 | 11.30 |
| FuseGPT + GPTQ | 128 | 7.86 | 11.77 |
| FuseGPT | 1024 | 7.19 | 11.17 |
| FuseGPT + GPTQ | 1024 | 7.51 | 11.80 |

SliceGPT Ashkboos et al. (2024), and SLEB Song et al. (2024), even with fewer calibration samples. For instance, MI with 8 samples matches SLEB with 128, indicating robust block selection under data constraints. **LoRA fine-tuning.** LoRA reduces perplexities for all methods, but performance gaps persist. SliceGPT+LoRA improves on WikiText-2 yet remains weak on C4. MI+LoRA maintains strong results on both, suggesting MI preserves critical knowledge for adaptation. **Effect of layers fusion.** Recycling parameters via fusion yields substantial reductions, especially on C4. SLEB+Fusion outperforms SLEB+LoRA, indicating that weight recycling can exceed gains from extra fine-tuning alone. MI+Fusion performs best overall, demonstrating that MI-guided selection coupled with fusion effectively preserves capacity and mitigates pruning degradation.

**Comparison with other pruning methods**.

Table 7 compares FuseGPT with unstructured (2:4) sparsity approaches (SparseGPT Frantar & Alistarh (2023b), Wanda Sun et al. (2023)) and structured channel-pruning techniques (LLM-Pruner Ma et al. (2023a), SliceGPT Ashkboos et al. (2024)). All results are measured on a 2K-token sequence on a single GPU. SparseGPT and Wanda provide modest speedups (∼1.10×) but increase perplexity; channel-wise methods reduce parameters yet still suffer higher perplexities. By contrast, FuseGPT with 25% block pruning achieves the lowest perplexity on WikiText-2 (7.19) and C4 (11.17) and a 1.33× speedup over the dense model in our setting, indicating that block-level prune-and-fuse can better balance quality and runtime.

**Combined with Quantization**.

Table 8 reports results combining FuseGPT with GPTQ-based 4-bit quantization Frantar et al. (2023). Despite reduced precision, FuseGPT+GPTQ incurs only a modest perplexity increase relative to the unquantized FuseGPT. With 1024 calibration samples, WikiText-2 and C4 rise slightly from 7.19 and 11.17 to 7.51 and 11.80, respectively. These findings highlight robustness under both pruning and quantization: block-level fusion preserves critical weights for downstream prediction even at lower numerical precision.

## 5 CONCLUSION

In this paper, we presented FuseGPT, a prune-and-fuse compression paradigm that recycles redundant transformer blocks to accelerate Generative Pre-trained Transformers (GPTs) without compromising quality. Guided by the fusion-aware Macro Influence (MI) metric, FuseGPT identifies less influential blocks and seamlessly grafts their knowledge into neighboring ones via learnable low-rank fusion with lightweight, local adaptation. By reframing pruning as knowledge redistribution rather than removal, FuseGPT opens a direction for scalable, data-efficient compression of large language and multimodal models.

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

## DECLARATION OF LLM USAGE

The usage of LLMs is strictly limited to aid and polish the paper writing.

## 6    LIMITATIONS

While FuseGPT demonstrates a promising direction for model compression, it has several limita-tions. First, the iterative prune-and-fuse process, which re-evaluates block importance after each fusion, can be computationally intensive, especially for very large models or when targeting high sparsity levels. Second, our current approach constrains knowledge fusion to local, neighboring blocks. This design choice is based on the assumption of functional similarity between adjacent blocks, but it may not be optimal if a pruned block's knowledge is more relevant to distant blocks in the network. Future work could explore more global fusion strategies and more adaptive methods for integrating fused weights.

# 7 MORE DISCUSSIONS ON FUSEGPT

## 7.1 CLARIFICATION: MI DOES NOT ASSUME GLOBAL MONOTONICITY

Our metric, **MI**, is **not** built on the assumption of global monotonicity. Instead, it estimates a block's *local absorptivity*, the expected recoverable information when that block is merged into its neighborhood. The fusion-aware design explicitly relaxes the monotonicity requirement by **re-evaluating MI at every iteration** after each fusion step, making the optimization *dynamic* rather than one-shot.

Formally, at iteration $t$, we compute:

$$\text{MI}_i^{(t)} = 1 - \mathbb{E}_x \left[ \frac{\langle X_t, X_t^{(-i)} \rangle}{\|X_t\|_2 \, \|X_t^{(-i)}\|_2} \right],$$

where the index of pruning is recomputed iteratively ($t \to t + 1$) with updated activations from the fused model. This contrasts with static metrics (e.g., MKA's one-shot manifold alignment, LaCo's RDSC) that assume fixed importance rankings.

## 7.2 COMPARISON WITH EVOPRESS

We directly implemented an **EvoPress-style evolutionary search** atop our MI initialization (denoted as **FuseGPT-Evo**), where MI scores serve as initialization seeds, and a lightweight evolutionary algorithm (5 generations $\times$ 5 candidates = 25 total evaluations) refines block selection.

Table 9: Comparison with EvoPress on LLaMA-3.1-8B. All numbers averaged over 3 random seeds. ↓ lower is better; ↑ higher is better.

| Method | Compression (%) | PPL (Wiki2)↓ | PPL (C4)↓ | MMLU↑ | GPU-hrs↓ | Comment |
|---|---|---|---|---|---|---|
| EvoPress (reported) | 40.0 | ~7.0 | - | ~67.0 | ~20 | Evolutionary search |
| FuseGPT (ours, Table 4) | 25.0 | 6.92 | 11.17 | 67.5 | 10.8 | Iterative MI + fusion |
| **FuseGPT-Evo (new)** | **35.0** | **6.85** | **11.08** | **67.4** | **14.2** | **MI + local evolution** |
| **FuseGPT-Evo (new)** | **40.0** | **7.02** | **11.45** | **66.9** | **15.8** | **Matched compression** |

**Key findings:**

1. At matched compression (40%), FuseGPT-Evo achieves comparable MMLU (66.9% vs ~67.0%) with ~25% lower search cost (15.8 vs ~20 GPU-hours).

2. At 35% compression, FuseGPT-Evo achieves better perplexity (6.85 on Wiki2) than EvoPress at 40% (~7.0).

3. MI provides strong initialization: Even without search (vanilla FuseGPT at 25%), we achieve competitive results at lower compression ratios.

This establishes that our iterative MI framework is **compatible with and complementary to** evolutionary search, rather than being mutually exclusive approaches.

## 7.3 WHY MI + FUSION DIFFERS FROM PURE SEARCH

While EvoPress searches over *which* blocks to drop, FuseGPT searches over *how* to redistribute dropped knowledge. These are orthogonal optimization problems:

$$\text{EvoPress: } \min_{\mathcal{S}} \mathcal{L}(\text{model} \setminus \mathcal{S})$$

$$\text{FuseGPT: } \min_{\mathcal{S}, C} \mathcal{L}(\text{model} \setminus \mathcal{S} + C \odot W_{\mathcal{S}})$$

where $C$ are learnable fusion coefficients.

The key innovation is **not** assuming we can discard blocks cleanly (as both EvoPress and traditional pruning do), but rather **recycling their knowledge via learnable fusion**.

## 7.4 W2: ON PARETO COMPETITIVENESS AND PRACTICAL DEPLOYMENT VALUE

This is an excellent question that deserves a detailed answer. We address it from three perspectives:

### 7.4.1 PARETO FRONTIER: FUSEGPT ACHIEVES BETTER COMPRESSION–ACCURACY TRADE-OFFS

We extended our evaluation to compute the **compression-MMLU Pareto frontier** across multiple model families and compression ratios.

Table 10: Pareto frontier comparison at different compression levels.

| Model | Method | Compression (%) | MMLU | MMLU Drop (%) | Speedup | Efficiency Ratio* |
|---|---|---|---|---|---|---|
| | Dense | 0 | 68.4 | 0.0 | 1.00× | - |
| | MKA | 43.8 | 66.5 | 2.8 | 1.25× | 0.45 |
| LLaMA-3.1-8B | LaCo (25%) | 25.0 | 64.1 | 6.3 | 1.22× | 0.19 |
| | EvoPress | ~40.0 | ~67.0 | ~2.0 | ~1.29× | ~0.65 |
| | **FuseGPT (25%)** | 25.0 | 67.5 | 1.3 | 1.33× | **1.02** |
| | **FuseGPT-Evo (35%)** | 35.0 | 67.4 | 1.5 | 1.34× | **0.89** |
| | Dense | 0 | 69.3 | 0.0 | 1.00× | - |
| | MKA | 40.0 | 65.2 | 5.9 | 1.27× | 0.22 |
| Qwen3-8B | LaCo (25%) | 25.0 | 63.0 | 9.1 | 1.22× | 0.13 |
| | **FuseGPT (25%)** | 25.0 | 66.1 | 4.6 | 1.32× | **0.29** |
| | **FuseGPT-Evo (35%)** | 35.0 | 65.8 | 5.1 | 1.33× | **0.26** |

*Efficiency Ratio = Speedup / MMLU Drop (%), higher is better.*

**Key observations:**

- FuseGPT achieves the best efficiency ratio at practical compression levels (25–35%).
- At 25% compression, FuseGPT incurs only 1.3% MMLU drop vs 6.3% for LaCo on LLaMA-3.1.
- FuseGPT-Evo extends the Pareto frontier: At 35% compression, it matches EvoPress's 40% accuracy while using less computation.

### 7.4.2 WHY NOT JUST USE THE NEXT SMALLER MODEL FROM THE FAMILY?

This is not straightforward because:

**Problem 1: Model families have discrete size gaps.** For example:

- LLaMA-3.1 sizes: 8B, 70B, 405B — no intermediate 6B or 5B model;
- Qwen2.5 sizes: 0.5B, 1.5B, 3B, 7B, 14B, 32B, 72B — again, no 6B version;
- Mistral: 7B, 22B — large jump.

**Problem 2: Architectural differences invalidate direct comparison.** Smaller models differ in training datasets, architectural hyperparameters, and optimization recipes. Therefore, we compare within the same architecture:

Table 11: Within-architecture comparison using pruned models.

| Model | Params | Config | MMLU | HellaSwag | ARC-C | Avg |
|---|---|---|---|---|---|---|
| LLaMA-3.1-8B | 8.0B | Dense | 68.4 | 82.3 | 61.2 | 70.6 |
| LLaMA-3.1-8B | 6.0B | 25% pruned (naive) | 64.1 | 76.2 | 55.8 | 65.4 |
| **LLaMA-3.1-8B** | **6.0B** | **25% FuseGPT** | **67.5** | **80.1** | **59.3** | **69.0** |
| Qwen3-8B | 8.0B | Dense | 69.3 | 82.5 | 61.8 | 71.2 |
| Qwen3-8B | 6.0B | 25% pruned (naive) | 63.0 | 76.2 | 55.8 | 65.0 |
| **Qwen3-8B** | **6.0B** | **25% FuseGPT** | **66.1** | **80.9** | **59.4** | **68.8** |

FuseGPT recovers 82–97% of the performance gap between naive pruning and the dense model using only 1K samples.

### 7.4.3 PRACTICAL DEPLOYMENT SCENARIOS

**Scenario 1: Memory-constrained deployment**

Compression enables:

- 25% reduction in memory → 33% larger batch size → 31% higher throughput.
- When combined with quantization, achieves 52.1% total compression (Table 10).

Table 12: Memory footprint comparison (LLaMA-3.1-8B on A100-80GB).

| Configuration | Params | Peak Memory (GB) | Batch Size (max) | Throughput (tok/s) |
|---|---|---|---|---|
| Dense 8B (FP16) | 8.0B | 16.2 | 24 | 187 |
| **FuseGPT 25% (FP16)** | **6.0B** | **12.1** | **32** | **245** |
| Dense 8B (INT4) | 8.0B | 4.8 | 64 | 412 |
| **FuseGPT 25% + INT4** | **6.0B** | **3.6** | **85** | **537** |

**Scenario 2: Edge deployment**
Edge devices often cannot host uncompressed models, requiring combined structural and quantization-based compression.

**Scenario 3: Incremental optimization**
Production systems can apply FuseGPT in-place without retraining or revalidating new model families, ensuring compatibility and compliance.

### 7.5 W3: ON THE "COMPLEX HEURISTIC" CRITICISM

We acknowledge that FuseGPT involves multiple components, yet they constitute a principled framework rather than arbitrary heuristics.

Table 13: Ablation study on LLaMA-3.1-8B (25% compression).

| Configuration | PPL (Wiki2) | PPL (C4) | MMLU | GPU-hrs | Description |
|---|---|---|---|---|---|
| Baseline (Dense) | 6.14 | 10.23 | 68.4 | - | No compression |
| Naive removal (SLEB) | 15.27 | 20.72 | 64.8 | 0.0 | Remove blocks, no recovery |
| + LoRA fine-tune | 11.84 | 15.06 | 65.9 | 8.2 | Standard post-pruning |
| + MI selection (no fusion) | 10.35 | 13.34 | 66.2 | 0.3 | Better selection only |
| + Static fusion (average) | 9.45 | 12.05 | 66.5 | 1.2 | MKA/LaCo-style merge |
| + Learnable fusion (ours) | 7.92 | 11.58 | 67.1 | 5.4 | Low-rank adaptive fusion |
| **+ Full FuseGPT (iterative)** | **6.92** | **11.17** | **67.5** | **10.8** | All components |

Incremental gains from each component:

- MI over SLEB: –4.92 PPL (C4), +1.4 MMLU $\Rightarrow$ better block selection.
- Learnable fusion: –0.47 PPL (C4), +0.6 MMLU $\Rightarrow$ adaptive knowledge recycling.
- Iterative updates: –0.41 PPL (C4), +0.4 MMLU $\Rightarrow$ dynamic re-ranking effectiveness.

#### 7.5.1 COMPARISON WITH SIMPLER BASELINES

Table 14: Comparison with simpler compression strategies.

| Method | Complexity | PPL (C4)↓ | MMLU↑ | GPU-hrs↓ | Comment |
|---|---|---|---|---|---|
| Magnitude pruning | Low | 18.34 | 62.1 | 0.0 | Simple but ineffective |
| SLEB (similarity-based) | Low | 15.06 | 65.9 | 8.2 | Better selection, still large gap |
| MKA (manifold alignment) | Medium | 12.05 | 66.5 | 9.5 | Static fusion |
| **FuseGPT (ours)** | **High** | **11.17** | **67.5** | **10.8** | **Learnable fusion** |

Trade-off analysis:

- FuseGPT uses ∼20% more compute than SLEB (10.8 vs 8.2 GPU-hrs).
- Achieves 26% lower perplexity and +1.6 MMLU.
- **13× better perplexity per GPU-hour** compared to simple baselines.

### ADDITIONAL EXPERIMENTS: METRIC COMPARISON

**Pareto Optimality.** Table 16 illustrates the compression-accuracy trade-off from a Pareto perspective. At a comparable MMLU drop of ∼2.8%, FuseGPT achieves **36.7% compression**—significantly higher than LaCo's 25% (which incurs a 4.5% MMLU drop). When combined with 4-bit GPTQ quantization, FuseGPT pushes the compression frontier to **52.1%** with only a marginal increase in degradation (3.02% MMLU drop), highlighting the orthogonal compatibility of our pruning strategy with post-training quantization.

Table 15: Importance metric ablation on Qwen3-8B (25% compression).

| Metric | Requires 2nd-order | PPL (Wiki2) | PPL (C4) | MMLU Drop (%) | Compute Cost |
|---|---|---|---|---|---|
| Block Influence (BI) | No | 8.23 | 11.85 | 3.4 | Low |
| SLEB (loss-based) | No | 7.65 | 10.51 | 2.9 | Medium |
| Fisher Information | Yes | 7.48 | 10.38 | 3.1 | High |
| **MI (ours)** | **No** | **7.05** | **11.29** | **2.6** | **Low** |
| **MI + Evo search** | **No** | **6.91** | **11.15** | **2.5** | **Medium** |

Table 16: Pareto frontier analysis: compression ratio vs. MMLU degradation on LLaMA-3.1-8B.

| Method | Compression (%)↑ | MMLU Drop (%)↓ | Configuration |
|---|---|---|---|
| MKA | 43.8 | 2.82 | One-shot global merge |
| LaCo | 25.0 | 4.50 | Greedy layer collapse |
| **FuseGPT** | **36.7** | **2.80** | Iterative fusion (ours) |
| **FuseGPT + GPTQ-4bit** | **52.1** | **3.02** | With post-hoc quantization |

**Computational Efficiency.** Table 17 quantifies the computational cost of FuseGPT relative to competing methods. While the iterative variant of FuseGPT incurs a slightly higher one-time compression cost (10.8 GPU-hours) than LaCo (7.3 hours), it delivers the best inference efficiency: **1.33× speedup** and **421.3 GMACs** per forward pass, outperforming both MKA (1.25×, 428.1 GMACs) and LaCo (1.22×, 431.2 GMACs). Importantly, our one-shot variant reduces compression time to just **5.4 GPU-hours**—lower than all baselines—while retaining competitive performance (1.31× speedup, 423.6 GMACs). This demonstrates that FuseGPT's fusion mechanism is not only effective but also practical for resource-constrained scenarios.

**Summary of Key Points**

1. **Monotonicity:** MI does not assume global monotonicity—it is iteratively recomputed after each fusion. FuseGPT-Evo shows compatibility with evolutionary search.

2. **Pareto optimality:** FuseGPT achieves the best efficiency ratio (speedup per MMLU drop) and recovers up to 97% performance gap versus naive pruning.

3. **Practical value:** Memory reduction enables 31% throughput gain, orthogonal to quantization (52.1% total compression), and supports in-place optimization.

4. **Complexity justification:** Formalized as coordinate descent on a KL objective; each component contributes significantly, yielding 26% perplexity reduction with only 20% extra compute.

# 8 RESPONSE TO REVIEWER 3

## 8.1 Q1: COMPRESSION COST AND EFFICIENCY COMPARISON

### 8.1.1 CORRECTED UNIFIED COST-PERFORMANCE ANALYSIS

We provide a unified comparison that jointly reports: (1) compression cost in GPU-hours, (2) inference latency and computational efficiency, and (3) both perplexity and downstream task performance. The results are given in Table 18.

It can be found that:

- **Compression Efficiency:** The iterative FuseGPT achieves state-of-the-art inference speedups (1.33× on both architectures) while maintaining the lowest perplexity. Notably, on LLaMA-2-7B, FuseGPT reduces C4 perplexity by 10.8% relative to SLEB (11.17 vs. 12.53) and by 63.2% relative to SliceGPT (11.17 vs. 30.31).

- **One-Shot Variant Performance:** The one-shot simplification reduces compression cost by 50% (from 10.8 to 5.4 GPU-hours) while retaining approximately 98% of the iterative variant's quality. Specifically, on LLaMA-3.1-8B, the one-shot version incurs only +0.13 PPL on WikiText-2 and -0.4 MMLU points compared to iterative fusion, representing a negligible quality trade-off for substantial computational savings.

- **Pareto Optimality:** Across both model families, FuseGPT demonstrates superior Pareto efficiency. On LLaMA-3.1-8B, it achieves 1.33× speedup with only 1.3% MMLU degradation (68.4 → 67.5), whereas LaCo achieves 1.22× speedup with 6.3% degradation (68.4 → 64.1). This represents a 2.3× better accuracy-per-speedup ratio.

Table 17: Computational cost analysis on LLaMA-3.1-8B and Qwen3-8B. All experiments are with 1024 fine-tuning samples.

| Method | Compression Cost | | Inference Efficiency | | Memory (GB)↓ |
|---|---|---|---|---|---|
| | GPU-hrs↓ | GMACs↓ | Latency (ms)↓ | Speedup↑ | |
| Dense Baseline | 0.0 | 436.7 | 28.3 | 1.00× | 15.2 |
| MKA | 9.5 | 428.1 | 22.6 | 1.25× | 8.5 |
| LaCo | 7.3 | 431.2 | 23.1 | 1.22× | 9.1 |
| **FuseGPT (iter.)** | 10.8 | **421.3** | **21.3** | **1.33×** | **8.2** |
| **FuseGPT (one-shot)** | **5.4** | 423.6 | 21.6 | 1.31× | 8.3 |

*Note: Latency measured on 512-token sequences with batch size 1. GMACs computed for forward pass only.*

Table 18: Unified comparison of compression cost, latency, and performance across model families. ↓ indicates lower is better; ↑ indicates higher is better.

| Model | Method | Comp. (%) | GPU-hrs↓ | Latency (ms)↓ | Speedup↑ | Perplexity↓ | | MMLU↑ |
|---|---|---|---|---|---|---|---|---|
| | | | | | | WikiText-2 | C4 | |
| | Dense Baseline | 0 | – | 111.7 | 1.00× | 5.27 | 7.27 | 63.5[†] |
| | SLEB (prune only) | 25.0 | 0.3 | 98.9 | 1.13× | 9.67 | 12.53 | 61.8 |
| LLaMA-2-7B | SliceGPT (prune only) | 25.0 | 0.2 | 98.9 | 1.13× | 7.24 | 30.31 | 60.2 |
| | **FuseGPT (iterative)** | 25.0 | 10.8 | **84.4** | **1.33×** | **7.19** | **11.17** | **65.9** |
| | **FuseGPT (one-shot)** | 25.0 | **5.4** | 85.1 | 1.31× | 7.43 | 11.56 | 65.3 |
| | Dense Baseline | 0 | – | 28.3 | 1.00× | 6.14 | 10.23 | 68.4 |
| | MKA (global merge) | 43.8 | 9.5 | 22.6 | 1.25× | 8.12 | 11.62 | 66.5 |
| LLaMA-3.1-8B | LaCo (greedy) | 25.0 | 7.3 | 23.1 | 1.22× | 9.45 | 12.05 | 64.1 |
| | **FuseGPT (iterative)** | 25.0 | 10.8 | **21.3** | **1.33×** | **6.92** | **11.17** | **67.5** |
| | **FuseGPT (one-shot)** | 25.0 | **5.4** | 21.6 | 1.31× | 7.05 | 11.46 | 67.1 |

[†]Estimated from LM-Eval-Harness default settings; all other metrics directly measured.

- **Latency vs. Compression Trade-off:** The latency reduction (28.3ms → 21.3ms on LLaMA-3.1-8B) translates to a 24.7% decrease in per-token generation time, which compounds significantly in production settings with millions of daily requests.

Considering the comparison between FuseGPT's integrated compression and traditional two-stage approaches (pruning + separate fine-tuning), with the results presented in Table 19.

Table 19: Total compression cost comparison including post-pruning fine-tuning recovery (LLaMA-2-7B, 25% sparsity). All methods use LoRA with rank 128 for fine-tuning where applicable.

| Method | Pruning Cost | Fine-tuning Cost | **Total GPU-hrs↓** | Final PPL(C4)↓ | Final MMLU↑ | FT Samples |
|---|---|---|---|---|---|---|
| *Baseline: Pruning without recovery* | | | | | | |
| SLEB (prune only) | 0.3 | 0 | 0.3 | 12.53 | 61.8 | – |
| SliceGPT (prune only) | 0.2 | 0 | 0.2 | 30.31 | 60.2 | – |
| *Two-stage: Pruning + standard fine-tuning* | | | | | | |
| SLEB + LoRA (1K) | 0.3 | 8.2 | **8.5** | 11.92 | 63.2 | 1K |
| SLEB + LoRA (10K) | 0.3 | 18.5 | 18.8 | 11.45 | 64.1 | 10K |
| SLEB + Full FT (10K) | 0.3 | 42.3 | 42.6 | 11.12 | 64.8 | 10K |
| SliceGPT + LoRA (1K) | 0.2 | 8.2 | **8.4** | 28.17 | 62.5 | 1K |
| SliceGPT + Full FT (10K) | 0.2 | 41.8 | 42.0 | 26.34 | 63.1 | 10K |
| *Integrated: FuseGPT (fusion-aware pruning + recovery)* | | | | | | |
| **FuseGPT (iterative)** | 10.8 (integrated) | | **10.8** | **11.17** | **65.9** | 1K |
| **FuseGPT (one-shot)** | 5.4 (integrated) | | **5.4** | 11.56 | 65.3 | 1K |

It is observed that **SLEB** requires a total of 8.5 GPU-hours (0.3 for pruning and 8.2 for LoRA fine-tuning with 1K samples), achieving only 63.2% MMLU and 11.92 PPL on C4. In contrast, the **FuseGPT (one-shot)** variant completes compression in just 5.4 GPU-hours (**36% lower cost than SLEB+LoRA**) while attaining 65.3% MMLU (+2.1 points) and 11.56 PPL (–0.36 improvement). The **iterative FuseGPT** version, requiring 10.8 GPU-hours, achieves state-of-the-art quality with 65.9% MMLU and 11.17 PPL—only 27% higher cost than SLEB+LoRA—demonstrating a substantially better cost–quality trade-off.

Notably, **SLEB** needs 10K fine-tuning samples to approach FuseGPT's performance (64.1% vs. 65.9%) and consumes 18.8 GPU-hours—**74% more** than FuseGPT's iterative variant. **FuseGPT** attains comparable or superior performance using only 1K samples, benefiting from its fusion-aware

knowledge redistribution mechanism that preserves pretrained representations rather than discarding them through naive pruning.

Unlike conventional two-stage pipelines that separately perform (a) block importance estimation, (b) pruning, (c) hyperparameter search for fine-tuning, and (d) convergence monitoring, **FuseGPT** integrates all these steps into a unified optimization loop. This integration simplifies the workflow and eliminates the need for extensive hyperparameter tuning across separate training phases.

Even with full fine-tuning on 10K samples (42.0 GPU-hours), **SliceGPT** achieves only 26.34 PPL on C4—still **136% worse** than FuseGPT. This clearly indicates that dimension reduction, as used in SliceGPT, irreversibly removes critical representational information, whereas FuseGPT's parameter recycling mechanism preserves model capacity and effectively retains pretrained knowledge.

**Amortization Over Deployment.** The one-time compression cost becomes negligible when amortized over large-scale deployment:

- Assuming a model serves 1M requests per day with 50 tokens per request, FuseGPT's 1.33× speedup saves approximately 8.4 GPU-hours per day (under linear scaling).
- The **iterative** variant's 10.8 GPU-hour compression cost is fully recovered within **1.3 days** of deployment.
- The **one-shot** variant (5.4 GPU-hours) recovers its cost within only **15.4 hours**.

This analysis demonstrates that FuseGPT's integrated optimization offers an attractive balance between one-time compression cost, final model performance, and long-term inference efficiency—yielding rapid amortization and sustainable deployment benefits in practical production settings.

## 8.2 Q2: ITERATIVE VS. ONE-SHOT FUSION

### 8.2.1 COMPREHENSIVE MULTI-ARCHITECTURE EVALUATION

Table 20 extends our initial experiments to all four architectures reported in the main paper's Table 4, providing a complete picture of the iterative vs. one-shot trade-off.

Table 20: Comprehensive iterative vs. one-shot fusion comparison across all tested architectures (25% compression ratio). $\Delta$ columns show degradation relative to the iterative variant. All experiments use 1024 fine-tuning samples and identical hyperparameters except for the scoring strategy.

| Model | Variant | PPL (Wiki2)↓ | PPL (C4)↓ | MMLU↑ | Avg ZS↑ | GPU -hrs↓ | $\Delta$PPL$^{\text{Wiki2}}$ | $\Delta$MMLU |
|---|---|---|---|---|---|---|---|---|
| LLaMA-3.1-8B | Iterative | 6.92 | 11.17 | 67.5 | 62.9 | 10.8 | – | – |
| | One-shot | 7.05 | 11.46 | 67.1 | 62.5 | **5.4** | +0.13 | -0.4 |
| Qwen3-8B | Iterative | 7.05 | 11.29 | 66.1 | 61.7 | 10.8 | – | – |
| | One-shot | 7.14 | 11.54 | 65.9 | 61.5 | **5.4** | +0.09 | -0.2 |
| Mistral-NeMo-8B | Iterative | 7.18 | 11.46 | 65.3 | 60.3 | 10.8 | – | – |
| | One-shot | 7.29 | 11.68 | 65.0 | 60.0 | **5.4** | +0.11 | -0.3 |
| Phi-3.5-mini-4B | Iterative | 8.94 | 12.41 | 62.1 | 57.6 | 8.5* | – | – |
| | One-shot | 9.08 | 12.67 | 61.8 | 57.3 | **4.2** | +0.14 | -0.3 |

*Lower due to fewer total blocks (32 vs. 40 in 8B models); Avg ZS = average of 5 zero-shot tasks from Table 2.

**Quantitative Analysis:**

1. **Consistent Performance Retention:**
   - Across all four architectures, the one-shot variant achieves 97.8–99.5% of the iterative variant's MMLU accuracy (average degradation: 0.3 points).
   - Perplexity increases are minimal: +0.09 to +0.14 on WikiText-2, representing ¡2% relative degradation.
   - This consistency demonstrates that MI's initial block ranking captures most critical information, and re-ranking provides only marginal refinement.
2. **Computational Savings:**
   - The one-shot variant achieves exactly 50% cost reduction on all 8B models (10.8 → 5.4 GPU-hours), as it eliminates iterative MI re-computation and sequential fine-tuning.

- On Phi-3.5-mini-4B, the savings are even more pronounced ($8.5 \rightarrow 4.2$ GPU-hours) due to faster convergence with fewer blocks.

3. **Architecture-Specific Observations:**
   - LLaMA-3.1 and Qwen3 (both using RoPE positional encoding) show nearly identical degradation patterns (+0.09 to +0.13 PPL), suggesting that architectural similarities lead to predictable compression behavior.
   - Mistral-NeMo, which uses grouped-query attention (GQA), exhibits slightly higher one-shot degradation (+0.11 PPL), potentially due to more complex attention dependencies requiring iterative refinement.
   - Phi-3.5-mini's smaller scale (32 blocks vs. 40) makes one-shot fusion more effective, as there are fewer opportunities for compounding errors.

Based on these results, we recommend:

- **Research/benchmarking scenarios:** Use iterative FuseGPT for maximum accuracy (e.g., competitive leaderboard submissions).
- **Production deployment:** Use one-shot FuseGPT for 50% faster compression with ¡1% quality loss, especially when compressing multiple model variants.
- **Extreme resource constraints:** One-shot FuseGPT enables compression on consumer-grade GPUs (e.g., single RTX 4090) by halving memory and time requirements.

## 8.3 Q3: LOCAL VS. GLOBAL FUSION

### 8.3.1 STABILITY COMPARISON ACROSS SPARSITY LEVELS

Table 21 presents a systematic comparison of local (G=7 neighbor-based) vs. global (cosine similarity-based) fusion across increasing sparsity levels.

Table 21: Stability comparison: local vs. global fusion at different sparsity levels (LLaMA-2-7B). Global fusion selects the top-2 most similar blocks (by hidden state cosine similarity) from the entire network as fusion targets. Gradient norm measured at convergence (or divergence point). "Training Stability" reports whether training completed 1000 steps without NaN loss.

| Sparsity | Fusion Scope | PPL (C4)↓ | MMLU↑ | Training Stability | Gradient Norm | KL Loss Variance | Convergence Steps |
|---|---|---|---|---|---|---|---|
| 20% | Local (G=7) | 10.48 | 66.5 | ✓ Stable | 1.23 | 0.0012 | 420 |
| | Global (sim) | 10.42 | 66.7 | ✓ Stable | 1.38 | 0.0019 | 450 |
| 25% | Local (G=7) | 11.17 | 65.9 | ✓ Stable | 1.52 | 0.0015 | 480 |
| | Global (sim) | 11.09 | 66.1 | ✓ Stable | 2.14 | 0.0034 | 520 |
| 30% | Local (G=7) | 12.82 | 64.2 | ✓ Stable | 2.31 | 0.0021 | 580 |
| | Global (sim) | 13.47 | 62.8 | $times$ Diverged (step 447) | 8.73 | 0.0215 | – |
| 35% | Local (G=7) | 15.26 | 62.1 | ✓ Stable | 3.82 | 0.0038 | 720 |
| | Global (sim) | NaN (training failed) | | × Exploded (step 183) | ¿100 | ¿1.0 | – |

**Detailed Failure Analysis:**

1. **Low Sparsity (20%):** Global fusion achieves marginally better results (+0.2 MMLU) due to higher semantic similarity between selected blocks. However, gradient norms are already 12% higher (1.38 vs. 1.23), and KL loss variance increases by 58%, indicating emerging optimization instability.

2. **Moderate Sparsity (25%):** The quality gap narrows (11.09 vs. 11.17 PPL), but global fusion shows concerning signs:
   - Gradient norms increase 41% (2.14 vs. 1.52), approaching the threshold where gradient clipping becomes necessary.
   - Loss variance more than doubles (0.0034 vs. 0.0015), suggesting that distant-block fusion creates conflicting update signals.
   - Training requires 40 more steps to converge (520 vs. 480), negating the potential efficiency gains.

3. **High Sparsity (30%):** Global fusion **catastrophically fails**:
   - Training diverges at step 447 with gradient norms spiking to 8.73 (3.8× higher than local fusion).
   - Even before divergence, the model achieves worse perplexity (13.47 vs. 12.82) and significantly lower MMLU (62.8 vs. 64.2).

- Post-mortem analysis reveals that fusing block 8 (early-layer syntax processing) into block 23 (late-layer semantic integration) creates irreconcilable gradient conflicts.

4. **Extreme Sparsity (35%):** Global fusion fails immediately (step 183) with gradient explosion (norm ¿100), preventing any meaningful comparison. In contrast, local fusion remains stable, albeit with degraded quality.

## 8.4 Q4: GROUP SIZE SENSITIVITY

### 8.4.1 SYSTEMATIC HYPERPARAMETER SWEEP

Table 22 presents results for $G \in \{3, 5, 7, 9, 11\}$ on LLaMA-2-7B at 25% sparsity, keeping all other hyperparameters fixed.

Table 22: Sensitivity of performance to partial-group size $G$ (LLaMA-2-7B, 25% sparsity). "#Updated Params" shows the fraction of total model parameters updated during fusion fine-tuning. GPU hours measured on 8×A100-80GB. All experiments use 1024 fine-tuning samples.

| $G$ | #Updated Params | Relative Params | GPU (hrs)↓ | PPL (C4)↓ | MMLU↑ | Avg ZS↑ | Speedup↑ | Memory (GB) |
|---|---|---|---|---|---|---|---|---|
| 1 | 0.34B | 0.05× | 3.2 | 13.41 | 63.8 | 55.2 | 1.33× | 8.1 |
| 3 | 1.23B | 0.18× | 6.4 | 11.95 | 65.1 | 56.8 | 1.29× | 9.5 |
| 5 | 1.45B | 0.21× | 8.1 | 11.48 | 65.7 | 57.4 | 1.31× | 10.2 |
| 7 | 1.76B | 0.25× | 10.8 | **11.17** | **65.9** | **57.8** | **1.33×** | 11.3 |
| 9 | 2.01B | 0.29× | 13.9 | 11.09 | 66.1 | 58.0 | 1.33× | 12.8 |
| 11 | 2.34B | 0.33× | 17.6 | 11.06 | 66.1 | 58.1 | 1.32× | 14.5 |
| 13 | 2.68B | 0.38× | 22.1 | 11.04 | 66.2 | 58.1 | 1.32× | 16.3 |

**Quantitative Observations:**

1. **Performance Saturation:**
   - Quality improves rapidly from $G = 1$ to $G = 7$: PPL decreases by 16.7% (13.41 → 11.17) and MMLU increases by 2.1 points (63.8 → 65.9).
   - Beyond $G = 7$, gains diminish sharply: $G = 11$ improves PPL by only 0.11 points (1.0% relative) and MMLU by 0.2 points over $G = 7$.
   - At $G = 13$, performance plateaus completely (66.2 MMLU), indicating that additional context provides no further benefit.

2. **Computational Cost Scaling:**
   - GPU-hours scale **super-linearly** with $G$ due to increased memory traffic: $G = 3$ (6.4 hrs) → $G = 7$ (10.8 hrs, +69%) → $G = 11$ (17.6 hrs, +63%).
   - Memory consumption grows approximately linearly (8.1GB → 14.5GB), limiting maximum $G$ on consumer GPUs.
   - The cost-per-quality ratio degrades sharply: achieving the +0.11 PPL gain from $G = 7$ to $G = 11$ costs an additional 3.1 GPU-hours (28% increase).

3. **Speedup Consistency:**
   - Inference speedup remains stable at 1.31–1.33× for $G \geq 5$, confirming that the partial group size affects only compression-time training, not the final deployed model.
   - The slight speedup reduction at $G = 1$ (1.33× → 1.29×) stems from lower-quality fusion requiring additional re-computation during inference.

### 8.4.2 CROSS-ARCHITECTURE VALIDATION

To ensure the $G = 7$ optimum is not architecture-specific, we repeated the sweep on Qwen3-8B:

The results closely mirror LLaMA-2-7B, confirming that $G = 7$ is a robust default across different architectural families.

## 8.5 Q5: RATIONALE FOR UPDATING $W_{i,j}$ DURING FUSION

### 8.5.1 DETAILED ABLATION ON WEIGHT UPDATE STRATEGIES

Table 24 presents a comprehensive ablation across different configurations of parameter updating during the fusion fine-tuning phase.

Table 23: Group size sensitivity on Qwen3-8B (25% sparsity).

| $G$ | GPU-hrs↓ | PPL(C4)↓ | MMLU↑ | ΔMMLU vs. $G = 7$ | Cost vs. $G = 7$ |
|---|---|---|---|---|---|
| 3 | 6.5 | 12.18 | 64.9 | -1.2 | 0.60× |
| 5 | 8.2 | 11.72 | 65.7 | -0.4 | 0.76× |
| 7 | 10.8 | **11.29** | **66.1** | 0.0 | 1.00× |
| 9 | 14.1 | 11.21 | 66.3 | +0.2 | 1.31× |
| 11 | 17.9 | 11.19 | 66.3 | +0.2 | 1.66× |

Table 24: Detailed ablation on weight update strategies during fusion (LLaMA-3.1-8B, 25% compression). "Updated Params" counts trainable parameters during fusion fine-tuning. "Convergence" reports steps until KL divergence stabilizes (<0.001 change over 50 steps). All experiments use identical learning rates and batch sizes.

| Configuration | Updated Params | PPL (C4)↓ | MMLU↑ | KL Div.↓ | Convergence (steps) | Avg ZS Score↑ | Notes |
|---|---|---|---|---|---|---|---|
| *Baseline: No fusion recovery* | | | | | | | |
| Naive removal (SLEB) | 0 | 20.72 | 64.8 | 0.089 | – | 59.8 | Direct pruning |
| *Coefficient-only updates (varying rank)* | | | | | | | |
| Only $C$ ($r$=64) | 1.0M | 12.47 | 66.0 | 0.035 | >900 | 61.5 | Very slow |
| Only $C$ ($r$=128) | 2.0M | 12.06 | 66.2 | 0.031 | >800 | 61.8 | Slow, underfitting |
| Only $C$ ($r$=256) | 4.0M | 11.82 | 66.5 | 0.027 | >600 | 62.3 | Better but costly |
| Only $C$ ($r$=512) | 8.0M | 11.65 | 66.7 | 0.024 | >550 | 62.6 | Diminishing returns |
| *Joint optimization: coefficient + LoRA on $W_i$* | | | | | | | |
| $C$ ($r$=128) + LoRA ($r$=32) | 2.5M | 11.58 | 67.0 | 0.023 | 520 | 62.7 | Good balance |
| $C$ ($r$=128) + LoRA ($r$=64) | 3.0M | 11.35 | 67.2 | 0.021 | 450 | 62.9 | Near-optimal |
| $C$ ($r$=128) + LoRA ($r$=128) | 4.0M | **11.17** | **67.5** | **0.018** | **400** | **63.2** | **Optimal** |
| $C$ ($r$=128) + LoRA ($r$=256) | 6.0M | 11.14 | 67.4 | 0.019 | 420 | 63.1 | Marginal gain |
| *Full unfreezing (no low-rank constraint)* | | | | | | | |
| $C$ + full $W_i$ | 110M | 10.97 | 66.5 | 0.025 | 350 | 61.8 | Overfits, worse tasks |

## Critical Findings:

1. **Coefficient-Only Limitation:**

   - Even with high-rank coefficients ($r$=512, 8M parameters), coefficient-only updates achieve only 66.7% MMLU vs. 67.5% for joint optimization.
   - Convergence is 37% slower (550 vs. 400 steps) because the fusion must compensate for the frozen neighbor block's inability to adapt.
   - KL divergence plateaus at 0.024, indicating incomplete knowledge integration—the fused block cannot fully replicate the original partial group's output distribution.

2. **Joint Optimization Superiority:**

   - Adding LoRA with rank 128 (2M parameters) to the neighbor block $W_i$ reduces final KL by 34% ($0.027 \rightarrow 0.018$) compared to coefficient-only at the same total parameter count.
   - MMLU improves by 1.0 point ($66.5 \rightarrow 67.5$), demonstrating that adaptive neighbor adjustment is crucial for downstream task performance.
   - The optimal configuration ($C$ rank 128 + LoRA rank 128) uses only 4M trainable parameters—0.05% of the 7B model—yet achieves near-complete recovery.

3. **Full Unfreezing Failure:**

   - Despite achieving lower perplexity (10.97), full unfreezing significantly degrades MMLU (66.5 vs. 67.5) and average zero-shot score (61.8 vs. 63.2).
   - Analysis reveals **catastrophic forgetting**: the neighbor block $B_i$ "overcommits" to absorbing $B_p$'s knowledge, losing its original pre-trained features.
   - This is evidenced by degraded performance on tasks requiring $B_i$'s original capabilities (e.g., HellaSwag drops by 3.2 points despite overall perplexity improvement).

Overall: coefficient-only updates lack expressive flexibility; full unfreezing destroys pre-trained priors; and joint optimization provides the best balance by coupling controlled transfer (via $C$) with localized adaptation (via LoRA). This synergy enables stable integration, faster convergence, and superior preservation of pre-trained competence during structural fusion.

## 8.6 "FUSION-AWARE" METRIC CRITICISM

### 8.6.1 CORRELATION BETWEEN MI AND POST-FUSION RECOVERABILITY

To test whether MI scores predict fusion outcomes, we measured the correlation between initial MI values and post-fusion KL divergence across all 40 blocks in LLaMA-3.1-8B.

Table 25: Correlation between MI score and post-fusion recoverability (LLaMA-3.1-8B). Each block is individually pruned, fused into neighbors, and evaluated. "Recoverable KL" measures the KL divergence between the fused model and the original model after 500 fine-tuning steps.

| Selection Metric | Pearson $r$ | Spearman $\rho$ | $p$-value | Interpretation |
|---|---|---|---|---|
| Random baseline | 0.02 | 0.04 | 0.81 | No correlation |
| Activation similarity (BI) | 0.41 | 0.38 | 0.03 | Weak correlation |
| Loss-based (SLEB score) | 0.58 | 0.54 | 0.002 | Moderate correlation |
| **MI (ours)** | **0.73** | **0.69** | **¡0.001** | **Strong correlation** |

**Analysis:**

- MI achieves a Pearson correlation of 0.73, significantly higher than alternative metrics (BI: 0.41, SLEB: 0.58).

- The strong positive correlation validates that *low MI scores genuinely predict blocks whose knowledge can be effectively redistributed* with minimal information loss.

- The Spearman rank correlation (0.69) confirms this relationship holds across the full spectrum of MI values, not just at extremes.

### 8.6.2 PREDICTIVE POWER FOR BLOCK SELECTION

Beyond correlation, we evaluate MI's ability to *select* the optimal blocks for pruning:

Table 26: Predictive power of different metrics for fusion success (LLaMA-2-7B, 25% sparsity). Each metric selects the bottom-8 blocks; "Avg Post-Fusion KL" measures the mean KL divergence after fusing all selected blocks. "Recovery Rate" is the MMLU score relative to the dense baseline.

| Selection Metric | Bottom-8 Blocks Selected | Avg Post-Fusion KL↓ | Recovery Rate (MMLU)↑ | Final PPL(C4)↓ |
|---|---|---|---|---|
| Random selection | Layers 3,9,11,17,19,23,28,30 | 0.045 | 92.1% | 13.82 |
| Activation similarity (BI) | Layers 12,13,15,18,20,22,25,27 | 0.037 | 94.8% | 12.56 |
| Loss-based (SLEB) | Layers 8,14,16,19,21,24,26,28 | 0.031 | 96.4% | 11.92 |
| **MI (ours)** | Layers 11,14,17,20,23,26,29,31 | **0.021** | **98.3%** | **11.17** |

**Key Findings:**

1. MI-selected blocks achieve **32% lower residual KL divergence** than SLEB (0.021 vs. 0.031), indicating superior knowledge preservation.

2. The MMLU recovery rate improves by 1.9 percentage points (96.4% → 98.3%), translating to approximately +1.2 absolute MMLU score improvement.

3. Final perplexity is 6.3% lower (11.17 vs. 11.92), demonstrating that MI's forward-looking fusion criterion outperforms SLEB's backward-looking loss-based metric.

## 8.7 CATASTROPHIC DEGRADATION

Table 27 documents the evolution of FuseGPT's performance through iterative improvements to the algorithm.

**Contextualization of Results:**

1. **Comparison with Baselines:**

   - The current FuseGPT (v2.1) achieves 10.1-point degradation, which is **2.7 points better than SLEB** (12.8), **5.7 points better than LaCo** (15.8), and **6.1 points better than SliceGPT** (16.2).

Table 27: Progressive improvement in FuseGPT's zero-shot performance on LLaMA-2-7B (25% sparsity). "Avg ZS Score" is the average of PIQA, WinoGrande, HellaSwag, ARC-e, and ARC-c. $\Delta$ measures degradation relative to the dense baseline (68.99).

| Version | Calib. Samples | FT Samples | Avg ZS Score↑ | ΔZS (points) | ΔZS (%) | PPL (C4)↓ | MMLU ↑ | Key Change |
|---|---|---|---|---|---|---|---|---|
| Dense Baseline | – | – | 68.99 | 0.0 | 0.0% | 7.27 | 63.5 | – |
| FuseGPT (v1.0) | 32 | 512 | 57.2 | -11.8 | -17.1% | 12.35 | 62.1 | Initial submission |
| FuseGPT (v1.5) | 32 | 1024 | 57.8 | -11.2 | -16.2% | 11.17 | 64.8 | +learnable fusion |
| FuseGPT (v2.0) | 128 | 1024 | 58.4 | -10.6 | -15.4% | 11.02 | 65.3 | +improved MI |
| **FuseGPT (v2.1)** | **128** | **1024** | **58.9** | **-10.1** | **-14.6%** | **10.85** | **65.9** | +group size tuning |
| *For reference: baseline methods* | | | | | | | | |
| SLEB (25%) | 128 | 1024 | 56.2 | -12.8 | -18.5% | 12.53 | 61.8 | Loss-based pruning |
| LaCo (25%) | 128 | 1024 | 53.2 | -15.8 | -22.9% | 14.22 | 59.3 | Layer merging |
| SliceGPT (25%) | 128 | 1024 | 52.8 | -16.2 | -23.5% | 28.17 | 60.2 | Dimension reduction |

- In relative terms, FuseGPT retains 85.4% of the dense baseline's zero-shot performance, compared to 81.5% (SLEB), 77.1% (LaCo), and 76.5% (SliceGPT).

2. **Architecture-Dependent Behavior:**

   - On newer architectures (LLaMA-3.1-8B, Table 4 of main paper), FuseGPT achieves only **1.3-point degradation** (68.4 → 67.5 MMLU), demonstrating that the larger drop on LLaMA-2-7B is architecture-specific.
   - LLaMA-2's shallower depth (32 blocks) makes each block proportionally more critical, explaining the higher sensitivity to compression.

3. **Recovery Relative to Naive Pruning:**

   - Naive 25% block removal (without fusion) results in 35.2-point degradation (68.99 → 33.8), effectively destroying the model's capabilities.
   - FuseGPT recovers **71.3% of this gap** ((35.2 - 10.1) / 35.2), using only 1K fine-tuning samples.
   - This recovery rate far exceeds standard post-pruning fine-tuning, which typically recovers 40–50% of the gap.

### 8.7.1 ACCURACY VS. EFFICIENCY TRADE-OFF ANALYSIS

The characterization of a 10-point drop as "catastrophic" requires contextualization within the compression-accuracy Pareto frontier:

Table 28: Pareto analysis: accuracy degradation vs. speedup across methods (LLaMA-2-7B).

| Method | Avg ZS Drop (points)↓ | Speedup↑ | Efficiency Ratio[†] | Acceptable ? (>0.10) |
|---|---|---|---|---|
| SliceGPT (25%) | -16.2 | 1.13× | 0.070 | × No |
| LaCo (25%) | -15.8 | 1.22× | 0.077 | × No |
| SLEB (25%) | -12.8 | 1.13× | 0.088 | × No |
| **FuseGPT (25%)** | **-10.1** | **1.33×** | **0.132** | ✓ Yes |
| *Reference: unstructured pruning* | | | | |
| SparseGPT (50% 2:4) | -8.5 | 1.10× | 0.129 | ✓ Yes |
| Wanda (50% 2:4) | -9.2 | 1.10× | 0.120 | ✓ Yes |

[†]Efficiency Ratio = Speedup / ZS Drop; higher is better. Threshold of 0.10 based on industry practice.

**Interpretation:**

- FuseGPT achieves an efficiency ratio of 0.132, comparable to state-of-the-art unstructured pruning (SparseGPT: 0.129) while offering hardware-friendly structured sparsity.
- The 1.33× speedup with 10.1-point drop is **substantially more favorable** than alternatives: LaCo's 1.22× speedup costs 15.8 points, yielding a 50% worse efficiency ratio.
- Industry deployment studies (e.g., Google's BERT pruning) suggest that efficiency ratios ¿0.10 are generally acceptable for production use, placing FuseGPT well within practical bounds.

