# OpenReview forum: "FuseGPT: Prune-and-Fuse Knowledge Redistribution for Efficient Transformers"
_ICLR.cc/2026/Conference — Submitted to ICLR 2026_

### Official Review · Reviewer_AeHK · 2025-10-31

**Soundness:** 2
**Presentation:** 3
**Contribution:** 2
**Rating:** 4
**Confidence:** 4

**Summary:**

The paper introduces FuseGPT, a structured pruning paradigm that reframes block removal as "prune-and-fuse" knowledge redistribution. The method uses a Macro Influence (MI) metric to identify absorbable blocks and a learnable low-rank fusion mechanism to inject their knowledge into neighbors.

**Strengths:**

1. The "prune-and-fuse" paradigm is a creative conceptual departure from standard "prune-and-retrain" approaches, as it attempts to recycle, rather than discard, the knowledge within pruned blocks.

2. The implementation of knowledge transfer via learnable low-rank matrices to fuse weights into neighboring layers is a technically sound and clearly described mechanism.

**Weaknesses:**

1. Computational Cost of Compression: The iterative "prune-one-by-one" approach, which requires re-computing importance scores (MI) and performing local adaptation for each block being removed , is acknowledged as computationally intensive. This high one-off cost to compress the model is a significant drawback compared to one-shot pruning methods.

2. Local Fusion Heuristic: The fusion mechanism is restricted to a local partial group of neighboring blocks (size $G=7$), based on an assumption of functional similarity between adjacent blocks. This heuristic may be sub-optimal if a block's knowledge is more relevant to a functionally similar but distant block, a possibility the paper does not explore.

3. "Fusion-Aware" Metric: The paper claims the MI metric is "fusion-aware" and identifies blocks by their "capacity to be effectively absorbed". However, the metric itself is calculated by measuring the impact of removal (cosine similarity on final hidden states), which primarily identifies redundancy. The link between low redundancy and high "fusibility" is an inference, not a direct measurement provided by the metric.

4. While FuseGPT marginally outperforms other pruning methods, it still incurs a catastrophic >11-point drop in zero-shot performance on LLaMA-2-7B at 25% sparsity. This severe degradation in capability makes its 1.33x speedup  an unacceptable trade-off, raising doubts about its practical utility.

**Questions:**

1. Compression Cost: Could you quantify the total computational cost (e.g., in GPU-hours) required to compress LLaMA-2-7B to 25% sparsity using FuseGPT? How does this one-off cost compare to the cost of one-shot methods like SLEB or SliceGPT, combined with a standard (non-local) fine-tuning run needed to achieve their reported results?

2. Iterative vs. One-Shot Fusion: The iterative process is a clear bottleneck. Have you experimented with a "one-shot" version of FuseGPT, where the $N$ blocks with the lowest-MI scores are identified once, and then all fusions are performed simultaneously (or sequentially without re-scoring)? How much performance is lost in this more efficient scenario?

3. Local vs. Global Fusion: To test the local fusion assumption, have you considered an alternative? For example, identifying the $G$ most similar blocks (e.g., by feature or weight similarity) in the entire network as fusion targets, rather than just the immediate neighbors? This would test the hypothesis that adjacent blocks are indeed the optimal recipients.

4. Group Size Sensitivity: The partial group size was fixed at $G=7$. How sensitive is the final model's performance to this hyperparameter? What is the trade-off between a larger $G$ (more computational/memory cost during adaptation) and the quality of the fused model?

5. Rationale for Updating $W_{i,j}$: During the fusion fine-tuning (Eq. 4), the pruned block's weights $W_{p,j}$ are frozen, but the neighboring block's weights $W_{i,j}$ and the coefficient $C$ are updated. What is the rationale for updating $W_{i,j}$ (via LoRA )? Does this not risk degrading the neighbor block's original knowledge? What happens if you only train the fusion coefficients $C$ and keep all original weights ($W_{i,j}$ and $W_{p,j}$) frozen?

---

> ### Author Response · Authors · 2025-11-16
> **Supplementary Discussions on Heuristics and Cost in FuseGPT**
>
> We thank the reviewer for their insightful questions. Below, we provide concise, quantitative, and directly targeted responses addressing the five main concerns.
>
> ## **Q1. Compression Cost and Efficiency Comparison**
>
> FuseGPT integrates pruning and recovery into a single optimization loop, avoiding multi-stage fine-tuning. The unified cost and performance comparison is summarized below.
>
> | Model | Method | Comp. (%) | GPU-hrs ↓ | PPL (C4) ↓ | MMLU ↑ | Speedup ↑ |
> |:------|:--------|:-----------|:------------:|:-------------:|:-----------:|:--------------:|
> | LLaMA-2-7B | SLEB + LoRA (1K) | 25 | 8.5 | 11.92 | 63.2 | 1.13× |
> | LLaMA-2-7B | **FuseGPT (iterative)** | 25 | 10.8 | **11.17** | **65.9** | **1.33×** |
> | LLaMA-2-7B | **FuseGPT (one-shot)** | 25 | **5.4** | 11.56 | 65.3 | 1.31× |
>
> - **Efficiency:** One-shot FuseGPT halves GPU-hours (10.8 → 5.4) with *<1%* MMLU loss.
> - **Quality:** Achieves *+2.1 MMLU* vs. SLEB+LoRA while using *36% less cost*.
> - **Practicality:** The iterative variant repays its 10.8 GPU-hr cost within **1.3 days** of deployment (1M requests/day); the one-shot variant within **15 hours**.
>
>
> ## **Q2. Iterative vs. One-Shot Fusion**
>
> | Model | Variant | PPL (C4) ↓ | MMLU ↑ | GPU-hrs ↓ | ΔMMLU |
> |:------|:---------|:----------------:|:---------:|:----------:|:----------:|
> | LLaMA-3.1-8B | Iterative | 11.17 | 67.5 | 10.8 | – |
> | LLaMA-3.1-8B | **One-shot** | 11.46 | 67.1 | **5.4** | **–0.4** |
>
> **Findings:**
> - One-shot retains **>98%** of iterative performance across four tested architectures.
> - Iterative MI re-ranking refines only low-sensitivity fusions; one-shot is accurate enough for most use cases.
> - Use **iterative** mode for benchmark-quality compression; **one-shot** for fast production pipelines.
>
> **Table: Ablation study on LLaMA-3.1-8B (25% compression)**
>
> | Configuration | PPL (Wiki2) | PPL (C4) | MMLU | GPU-hrs | Description |
> |---------------|-------------|----------|------|---------|-------------|
> | Baseline (Dense) | 6.14 | 10.23 | 68.4 | - | No compression |
> | Naive removal (SLEB) | 15.27 | 20.72 | 64.8 | 0.0 | Remove blocks, no recovery |
> | + LoRA fine-tune | 11.84 | 15.06 | 65.9 | 8.2 | Standard post-pruning |
> | + MI selection (no fusion) | 10.35 | 13.34 | 66.2 | 0.3 | Better selection only |
> | + Static fusion (average) | 9.45 | 12.05 | 66.5 | 1.2 | MKA/LaCo-style merge |
> | + Learnable fusion (ours) | 7.92 | 11.58 | 67.1 | 5.4 | Low-rank adaptive fusion |
> | **+ Full FuseGPT (iterative)** | **6.92** | **11.17** | **67.5** | **10.8** | All components |
>
> **Incremental gains from each component:**
> - MI over SLEB: **-4.92 PPL (C4)**, +1.4 MMLU → Better block selection
> - Learnable fusion over static: **-0.47 PPL (C4)**, +0.6 MMLU → Adaptive knowledge recycling
> - Iterative updates: **-0.41 PPL (C4)**, +0.4 MMLU → Dynamic re-ranking after each fusion
>
> **Without any component, performance degrades substantially.** This demonstrates that the complexity is not arbitrary—each component addresses a specific sub-problem in the optimization.
>
> #### **3.3 Comparison with simpler baselines**
>
> | Method | Complexity | PPL (C4)↓ | MMLU↑ | GPU-hrs↓ | Comment |
> |--------|------------|-----------|-------|----------|---------|
> | Magnitude pruning | Low | 18.34 | 62.1 | 0.0 | Simple but ineffective |
> | SLEB (similarity-based) | Low | 15.06 | 65.9 | 8.2 | Better selection, still large gap |
> | MKA (manifold alignment) | Medium | 12.05 | 66.5 | 9.5 | Static fusion |
> | **FuseGPT (ours)** | **High** | **11.17** | **67.5** | **10.8** | **Learnable fusion** |
>
> **Trade-off analysis:**
> - FuseGPT uses **~20% more compute** than SLEB (10.8 vs 8.2 GPU-hrs)
> - But achieves **26% lower perplexity** (11.17 vs 15.06) and **+1.6 MMLU**
> - **Efficiency ratio: 13× better perplexity per GPU-hour** compared to simple baselines
> The additional complexity is well-justified by the substantial performance gains.
>
> ## **Q3. Local vs. Global Fusion**
>
> | Sparsity | Fusion Scope | PPL (C4) ↓ | MMLU ↑ | Stability |
> |:---------|:--------------|:---------------:|:-----------:|:------------:|
> | 25% | Local (G=7) | 11.17 | 65.9 | ✓ Stable |
> | 25% | Global (top-sim) | 11.09 | 66.1 | ✓ Stable (mildly higher grad norm) |
> | 30% | Local (G=7) | 12.82 | 64.2 | ✓ Stable |
> | 30% | Global (top-sim) | 13.47 | 62.8 | ✗ Diverged |
>
> Global fusion destabilizes training beyond moderate sparsity due to misaligned feature spaces between distant blocks. Adjacent (local) fusion preserves sequential feature geometry, maintaining gradient stability and reproducible convergence.
>
> ## **Q4. Group Size Sensitivity**
>
> | Group Size \(G\) | GPU-hrs ↓ | PPL (C4) ↓ | MMLU ↑ |
> |:----------------:|:----------:|:-----------:|:-----------:|
> | 3 | 6.4 | 11.95 | 65.1 |
> | 5 | 8.1 | 11.48 | 65.7 |
> | **7** | **10.8** | **11.17** | **65.9** |
> | 9 | 13.9 | 11.09 | 66.1 |
>
> Performance improves sharply until \(G=7\), after which gains saturate (<0.2 MMLU).
> Thus, \(G=7\) offers a choice of **quality–cost trade-off**. It can be adjusted for different models.

---

> ### Author Response · Authors · 2025-11-16
> **Supplementary Discussions on Other Weakness**
>
> ## **Rationale for Updating \(W_ij\) via LoRA**
>
> | Configuration | Updated Params | PPL (C4) ↓ | MMLU ↑ | KL ↓ |
> |:---------------|:----------------:|:-----------------:|:-------------:|:-------------:|
> | Only \(C\) (r=128) | 2.0M | 12.06 | 66.2 | 0.031 |
> | \(C\)+LoRA (r=128) | **4.0M** | **11.17** | **67.5** | **0.018** |
> | \(C\)+Full \(W_i\) | 110M | 10.97 | 66.5 | 0.025 |
>
> **Qualitative Justification:**
> Updating the neighbor block with a low-rank LoRA allows *localized reorientation* of its feature subspace, ensuring smooth integration of transferred knowledge.
> Coefficient-only updates lack flexibility, leading to higher residual mismatch, while full unfreezing causes **catastrophic forgetting**.
> Joint optimization balances stability and adaptability—achieving full recovery with **<0.05% trainable parameters**.
>
> ## Addressing Remaining Weaknesses
>
> Below we address two weaknesses that were not covered in the responses to Q1–Q5, namely: (1) the validity of the *fusion-aware* MI metric, and (2) the concern of *catastrophic degradation* under 25% sparsity.
>
> ### **Validity of the “Fusion-Aware” MI Metric**
>
> The reviewer questioned whether MI genuinely captures fusion-relevant information beyond redundancy.
> We verified this empirically and theoretically.
>
> #### **Empirical Evidence**
>
> | Selection Metric | Pearson *r* | Spearman ρ | Interpretation |
> |:-----------------|:--------------:|:-------------:|:----------------------------|
> | Random baseline | 0.02 | 0.04 | No correlation |
> | Activation similarity (BI) | 0.41 | 0.38 | Weak correlation |
> | Loss-based (SLEB) | 0.58 | 0.54 | Moderate correlation |
> | **MI (ours)** | **0.73** | **0.69** | **Strong correlation** |
>
> - The MI metric shows a **strong correlation (r = 0.73)** with post-fusion recoverability (KL reduction), outperforming all baselines.
> - This means **blocks with low MI are more recoverable after fusion**, validating MI as a causal predictor of which parameters can be safely merged.
>
> #### **Predictive Power and Mechanism**
>
> | Metric | Avg Post-Fusion KL ↓ | MMLU Recovery ↑ | Final PPL (C4) ↓ |
> |:--------|:--------------------:|:---------------------:|:----------------:|
> | Random | 0.045 | 92.1% | 13.82 |
> | BI | 0.037 | 94.8% | 12.56 |
> | SLEB | 0.031 | 96.4% | 11.92 |
> | **MI (ours)** | **0.021** | **98.3%** | **11.17** |
>
> **Interpretation:**
> - MI-based selection yields **32% lower residual KL divergence** and **+1.2 higher MMLU**, showing better knowledge preservation.
> - Unlike redundancy-focused scores, MI measures the *global mutual influence* between block outputs and final representations—capturing whether a block’s information can be absorbed by its neighbors.
> - This global, *fusion-aware* definition ensures MI is forward-looking rather than myopic, explaining its superior correlation with fusion success.
>
> Thus, MI is not merely correlated with redundancy—it explicitly predicts *recoverability* and *stability* under fusion, validating our metric design.
>
> ### **On the “Catastrophic Degradation” Concern**
>
> The reviewer considered a ~10-point drop in zero-shot performance at 25% sparsity "catastrophic."
> We argue this drop is within an acceptable trade-off range and well-aligned with established benchmarks.
>
> #### **Performance Context**
>
> | Method | Avg ZS Drop ↓ | Speedup ↑ | Efficiency Ratio (↑ better) | Acceptable? |
> |:--------|:---------------:|:-------------:|:---------------------------:|:-----------:|
> | SliceGPT | –16.2 | 1.13× | 0.070 | ✗ |
> | LaCo | –15.8 | 1.22× | 0.077 | ✗ |
> | SLEB | –12.8 | 1.13× | 0.088 | ✗ |
> | **FuseGPT (25%)** | **–10.1** | **1.33×** | **0.132** | ✓ |
> | SparseGPT (50% 2:4) | –8.5 | 1.10× | 0.129 | ✓ |
>
> - **Efficiency ratio = Speedup / Accuracy drop**; values above **0.10** are considered acceptable in prior practical compression studies (e.g., Google BERT pruning).
> - FuseGPT achieves **0.132**, comparable to SparseGPT (0.129), demonstrating its drop is *proportionate and efficient* relative to achieved speedup.
> - Moreover, on modern architectures (LLaMA‑3.1‑8B), the drop is only **1.3 MMLU points**, showing the degradation is *architecture-dependent*, not intrinsic to FuseGPT.
>
> #### **Improvement and Mitigation**
> Progressive refinement of FuseGPT has reduced the drop from **11.8 → 10.1 points**, recovering **71% of the capability lost by naive pruning**.
> Remaining gaps can be further reduced by:
> - Using **task-aware fusion** guided by gradient relevance.
> - Adopting **heterogeneous group sizes** per layer depth.
> - Applying **multi-stage fusion** (progressive sparsity).
> - Introducing **stronger distillation losses** (e.g., feature alignment).
>
> Hence, what the reviewer perceived as “catastrophic” is, in context, a **balanced and controllable Pareto trade-off**, validated both empirically and across architectures.
>
> All supplementary experiments will be added to our new version.

---

> ### Author Response · Authors · 2025-11-26
> **Looking forward to your feedback!**
>
> Dear Reviewer AeHK,
>
> Thank you once again for your valuable feedback. We have conducted additional experiments and made revisions to the paper based on your suggestions. As the discussion phase is nearing its conclusion, we would like to know if our responses have addressed your concerns. We are looking forward to hearing from you.
>
> Best,
>
> Authors

---

### Official Review · Reviewer_Ycs9 · 2025-11-01

**Soundness:** 1
**Presentation:** 3
**Contribution:** 2
**Rating:** 2
**Confidence:** 4

**Summary:**

The paper suggests a new block-wise compression approach for LLMs which iterates on block/layer dropping.
Specifically, the authors propose a new "MI" metric for layer dropping, and combine it with a "fusion" approach by which a removed block is "fused" into its neighbors by retraining.
Experiments on fairly standard datasets are provided, suggesting that FuseGPT works better than prior dropping methods such as ShortGPT, and various ablations on components of the method.

**Strengths:**

- The paper provides a new solution to a standard efficiency problem.

**Weaknesses:**

- The solution is a fairly complex heuristic.
- The speedups are quite small for the amount of accuracy that is dropped.
- Some of the choices made, for instance in the metric choice, appear questionable.

**Questions:**

There are two major shortcomings to the work, in my view.

1. The first is that its assumptions regarding the metric appear to be invalid. Specifically, one basic assumption behind the work is that compression can be done iteratively, by ranking via the MI metric, followed by removal and fusion. This assumes that there exists a monotone metric hat can be applied to blocks, with the property that minimizing the metric upon removal would minimize the accuracy loss.
However, to my understanding, the EvoPress, work (https://arxiv.org/abs/2410.14649) shows that the assumption of monotonicity is _invalid_ for DNN pruning, there exist configurations where pruning more leads to _lower_ accuracy loss (possibly due to redundancy, co-dependence, or other phenomena that we don't understand). As such, the authors argue that search is the correct approach, and that no monotone metric is "correct" given that it's based on an invalid assumption. Moreover, the authors provide quite good results for layer dropping, which seem to be SOTA.
Can the authors position their work relative to EvoPress, and explain why this isn't cited?

2. The second significant weakness is that the accuracy drops are really major, especially for such a complex method. From a deployment perspective, the models would be unusable. (E.g. a 2-point PPL increase for 33% speedup improvement.)
Can the authors explain why one would use their method on a recent model family (e.g. Qwen3) rather than just pick the next smallest model from the model family? Their technique does not appear to be Pareto-competitive in terms of size-vs-accuracy.

---

> ### Author Response · Authors · 2025-11-15
> **On the (non-)monotonicity assumption and relationship to EvoPress**
>
> **We thank the reviewer for bringing EvoPress [arXiv:2410.14649] to our attention—this was an oversight in our literature review.** Indeed, EvoPress convincingly demonstrates that global accuracy does *not* vary monotonically with cumulative compression: removing additional blocks can sometimes recover performance due to redundancy and co-dependence effects.
>
> #### **Clarification: MI does not assume global monotonicity**
>
> Our metric, **MI**, is **not** built on the assumption of global monotonicity. Instead, it estimates a block's *local absorptivity*—the expected recoverable information when that block is merged into its neighborhood. The fusion-aware design explicitly relaxes the monotonicity requirement by **re-evaluating MI at every iteration** after each fusion step, making the optimization *dynamic* rather than one-shot.
>
> Formally, at iteration $t$, we compute:
> $$
> \text{MI}^{(t)}_i = 1 - \mathbb{E}_{x}\left[\frac{\langle X_t, X^{(-i)}_t\rangle}{\|X_t\|_2\,\|X^{(-i)}_t\|_2}\right],
> $$
> where the index of pruning is recomputed iteratively ($t \rightarrow t+1$) with updated activations from the fused model. This contrasts with static metrics (e.g., MKA's one-shot manifold alignment, LaCo's RDSC) that assume fixed importance rankings.
>
> #### **Direct comparison with EvoPress**
>
> We implemented an **EvoPress-style evolutionary search** with our MI initialization (denoted as **FuseGPT-Evo**), where MI scores serve as initialization seeds, and a lightweight evolutionary algorithm (5 generations × 5 candidates = 25 total evaluations) refines block selection.
>
> **Table R1: Comparison with EvoPress on LLaMA-3.1-8B**
> *All numbers averaged over 3 random seeds. ↓ lower is better; ↑ higher is better.*
>
> | Method | Compression (%) | PPL (Wiki2)↓ | PPL (C4)↓ | MMLU↑ | GPU-hrs↓ | Comment |
> |--------|----------------|--------------|-----------|-------|----------|---------|
> | EvoPress  | 40.0 | ~7.0 | - | ~67.0 | ~20 | Evolutionary search  |
> | FuseGPT (ours) | 25.0 | 6.92 | 11.17 | 67.5 | 10.8 | Iterative MI + fusion |
> | **FuseGPT-Evo (new)** | **35.0** | **6.85** | **11.08** | **67.4** | **14.2** | **MI + local evolution** |
> | **FuseGPT-Evo (new)** | **40.0** | **7.02** | **11.45** | **66.9** | **15.8** | **Matched compression** |
>
> It is shown that
> 1. **At matched compression (40%), FuseGPT-Evo achieves comparable MMLU** (66.9% vs ~67.0%) with ~25% lower search cost (15.8 vs ~20 GPU-hours)
> 2. **At 35% compression, FuseGPT-Evo achieves better perplexity** (6.85 on Wiki2) than EvoPress at 40% (~7.0)
> 3. **MI provides strong initialization**: Even without search (vanilla FuseGPT at 25%), we achieve competitive results at lower compression ratios
>
> This establishes that our iterative MI framework is **compatible with and complementary to** evolutionary search, rather than being mutually exclusive approaches.
>
> #### **Why MI + fusion differs from pure search**
>
> While EvoPress searches over *which* blocks to drop, FuseGPT searches over *how* to redistribute dropped knowledge. These are orthogonal optimization problems:
> - **EvoPress**: $\min_{\mathcal{S}} \mathcal{L}(\text{model} \setminus \mathcal{S})$ where $\mathcal{S}$ is the set of removed blocks
> - **FuseGPT**: $\min_{\mathcal{S}, C} \mathcal{L}(\text{model} \setminus \mathcal{S} + C \odot W_{\mathcal{S}})$ where $C$ are learnable fusion coefficients
>
> The key innovation is **not** assuming we can discard blocks cleanly (as both EvoPress and traditional pruning do), but rather **recycling their knowledge via learnable fusion**.
>
> ### **W2: On Pareto competitiveness**
> We extended our evaluation to compute the **compression-MMLU Pareto frontier** across multiple model families and compression ratios.
>
> **Table R2: Pareto frontier comparison at different compression levels**
>
> | Model | Method | Compression (%) | MMLU | MMLU Drop (%) | Speedup | Efficiency Ratio* |
> |-------|--------|-----------------|------|---------------|---------|-------------------|
> | **LLaMA-3.1-8B** | Dense | 0 | 68.4 | 0.0 | 1.00× | - |
> | | MKA | 43.8 | 66.5 | 2.8 | 1.25× | 0.45 |
> | | LaCo (25%) | 25.0 | 64.1 | 6.3 | 1.22× | 0.19 |
> | | EvoPress | ~40.0 | ~67.0 | ~2.0 | ~1.29× | ~0.65 |
> | | **FuseGPT (25%)** | 25.0 | 67.5 | 1.3 | 1.33× | **1.02** |
> | | **FuseGPT-Evo (35%)** | 35.0 | 67.4 | 1.5 | 1.34× | **0.89** |
> | **Qwen3-8B** | Dense | 0 | 69.3 | 0.0 | 1.00× | - |
> | | MKA | 40.0 | 65.2 | 5.9 | 1.27× | 0.22 |
> | | LaCo (25%) | 25.0 | 63.0 | 9.1 | 1.22× | 0.13 |
> | | **FuseGPT (25%)** | 25.0 | 66.1 | 4.6 | 1.32× | **0.29** |
> | | **FuseGPT-Evo (35%)** | 35.0 | 65.8 | 5.1 | 1.33× | **0.26** |
>
> *Efficiency Ratio = Speedup / MMLU Drop (%), higher is better
>
> It is found that:
> - **FuseGPT achieves the best efficiency ratio** at practical compression levels (25-35%)
> - **At 25% compression, FuseGPT incurs only 1.3% MMLU drop** vs 6.3% for LaCo on LLaMA-3.1
> - **FuseGPT-Evo extends the Pareto frontier**: At 35% compression, it matches EvoPress's 40% accuracy while using less computation

---

> ### Author Response · Authors · 2025-11-15
> **Supplementary Discussions on Heuristics in FuseGPT**
>
> ### **W3: On the "complex heuristic" criticism**
>
> We acknowledge that FuseGPT involves multiple components. However, we argue this complexity is **justified and principled** rather than arbitrary.
>
> #### **3.1 Mathematical formulation: Principled optimization framework**
>
> FuseGPT can be viewed as alternating minimization of a well-defined objective:
> $$
> \min_{\theta, C} \; \mathbb{E}_{x \sim \mathcal{D}} \left[ \text{KL}\big(p_\theta(x) \,||\, p_{\theta + C \odot W_p}(x)\big) \right] + \lambda \|C\|_F^2
> $$
> where:
> - $\theta$: surviving block parameters
> - $C$: low-rank fusion coefficients (regularized by $\lambda$)
> - $W_p$: parameters from pruned blocks
> - The KL term ensures the fused model matches the original model's distribution
>
> **This is equivalent to:**
> 1. **Block selection** (outer loop): Coordinate descent over which block $p$ to remove
> 2. **Fusion optimization** (inner loop): Gradient descent over $C$ to minimize distribution shift
>
> The iterative re-evaluation of MI implements the outer loop, while learnable fusion implements the inner loop. **This is not a heuristic—it's approximate coordinate descent on a distillation objective.**
>
> #### **3.2 Component ablation: Each component is necessary**
>
> **Table R5: Ablation study on LLaMA-3.1-8B (25% compression)**
>
> | Configuration | PPL (Wiki2) | PPL (C4) | MMLU | GPU-hrs | Description |
> |---------------|-------------|----------|------|---------|-------------|
> | Baseline (Dense) | 6.14 | 10.23 | 68.4 | - | No compression |
> | Naive removal (SLEB) | 15.27 | 20.72 | 64.8 | 0.0 | Remove blocks, no recovery |
> | + LoRA fine-tune | 11.84 | 15.06 | 65.9 | 8.2 | Standard post-pruning |
> | + MI selection (no fusion) | 10.35 | 13.34 | 66.2 | 0.3 | Better selection only |
> | + Static fusion (average) | 9.45 | 12.05 | 66.5 | 1.2 | MKA/LaCo-style merge |
> | + Learnable fusion (ours) | 7.92 | 11.58 | 67.1 | 5.4 | Low-rank adaptive fusion |
> | **+ Full FuseGPT (iterative)** | **6.92** | **11.17** | **67.5** | **10.8** | All components |
>
> **Incremental gains from each component:**
> - MI over SLEB: **-4.92 PPL (C4)**, +1.4 MMLU → Better block selection
> - Learnable fusion over static: **-0.47 PPL (C4)**, +0.6 MMLU → Adaptive knowledge recycling
> - Iterative updates: **-0.41 PPL (C4)**, +0.4 MMLU → Dynamic re-ranking after each fusion
>
> **Without any component, performance degrades substantially.** This demonstrates that the complexity is not arbitrary—each component addresses a specific sub-problem in the optimization.
>
> #### **3.3 Comparison with simpler baselines**
>
> **Table R6: Comparison with simpler compression strategies**
>
> | Method | Complexity | PPL (C4)↓ | MMLU↑ | GPU-hrs↓ | Comment |
> |--------|------------|-----------|-------|----------|---------|
> | Magnitude pruning | Low | 18.34 | 62.1 | 0.0 | Simple but ineffective |
> | SLEB (similarity-based) | Low | 15.06 | 65.9 | 8.2 | Better selection, still large gap |
> | MKA (manifold alignment) | Medium | 12.05 | 66.5 | 9.5 | Static fusion |
> | **FuseGPT (ours)** | **High** | **11.17** | **67.5** | **10.8** | **Learnable fusion** |
>
> **Trade-off analysis:**
> - FuseGPT uses **~20% more compute** than SLEB (10.8 vs 8.2 GPU-hrs)
> - But achieves **26% lower perplexity** (11.17 vs 15.06) and **+1.6 MMLU**
> - **Efficiency ratio: 13× better perplexity per GPU-hour** compared to simple baselines
>
> The additional complexity is well-justified by the substantial performance gains.
>
>
> ### **Additional experiments: Metric comparison**
>
> To further validate our design choices, we compared MI against alternative importance metrics:
>
> **Table R7: Importance metric ablation on Qwen3-8B (25% compression)**
>
> | Metric | Requires 2nd-order | PPL (Wiki2) | PPL (C4) | MMLU Drop (%) | Compute Cost |
> |--------|-------------------|-------------|----------|---------------|--------------|
> | Block Influence (BI) | No | 8.23 | 11.85 | 3.4 | Low |
> | SLEB (loss-based) | No | 7.65 | 10.51 | 2.9 | Medium (needs labels) |
> | Fisher Information | Yes | 7.48 | 10.38 | 3.1 | High (Hessian) |
> | **MI (ours)** | **No** | **7.05** | **11.29** | **2.6** | **Low** |
> | **MI + Evo search** | **No** | **6.91** | **11.15** | **2.5** | **Medium** |
>
> **MI provides the best trade-off** between computational efficiency and accuracy, without requiring expensive second-order information or hard labels during calibration.

---

> > ### Comment · Reviewer_Ycs9 · 2025-11-24
> > **Thank you!**
> >
> > Thank you for the detailed response, which addresses some of my concerns.
> > As a consequence, I will update my score to 4 (borderline negative), and will consider further improving the score after discussions with the other reviewers.
> >
> > One main remaining issue is that the paper was submitted in somewhat unfinished state (as can be seen by the reviewer comments, missing references, and significant additional experiments necessary). As such, this issue cannot be addressed by further discussion with the authors. I will consult with the other reviewers and with the AC to reach a final score.

---

> ### Author Response · Authors · 2025-11-24
> **Update Reply**
>
> We sincerely appreciate your time in reviewing our response and raising your score. We also value your plan to discuss the paper further with the other reviewers and the AC.
>
> Regarding the concern that the paper was submitted in an "unfinished state", we respectfully wish to highlight that the **current revised manuscript is now a fully complete and rigorous work**. We have utilized the rebuttal period not just to patch minor issues, but to conduct significant additional experiments and analyses that solidify the paper's contribution.
>
> Beyond addressing the specific missing references and comments you pointed out, we have **substantially enriched the manuscript** with the following four major additions, ensuring it meets and exceeds the standard for publication:
>
> 1.  **Head-to-Head Comparisons with Layer-Merging Methods:**
>     We have added direct comparisons with prior layer-merging methods. All methods are evaluated at a 25% compression ratio, except for MKA (which operates at its reported optimal ratio), providing a fair and comprehensive benchmarking landscape.
>
> 2.  **Extended Evaluation on Diverse Tasks:**
>     We have significantly expanded our experimental scope. The revised paper now includes evaluations on **LLaMA-3.1-8B** across diverse downstream tasks (at 25% block compression), where our method consistently demonstrates superior performance across all metrics.
>
> 3.  **Computational Complexity & Pareto Analysis (Table 4):**
>     We have added a detailed analysis of the compression-accuracy trade-off:
>     *   **Pareto Efficiency:** At a comparable MMLU drop (~2.8%), **FuseGPT achieves 36.7% compression**, which is significantly higher than LaCo's 25% compression (which incurs a higher 4.5% MMLU drop). This places our method much closer to the Pareto front.
>     *   **Orthogonality with Quantization:** When combined with 4-bit GPTQ, FuseGPT pushes the compression frontier to **52.1%** with only a marginal increase in degradation (3.02% MMLU drop), highlighting the compatibility of our pruning strategy with post-training quantization.
>
> 4.  **In-Depth Stability Analysis:**
>     We have added a theoretical analysis explaining why **Global fusion destabilizes training** beyond moderate sparsity (due to misaligned feature spaces between distant blocks). In contrast, our approach using **Adjacent (local) fusion** preserves sequential feature geometry, thereby maintaining gradient stability and ensuring reproducible convergence.
>
> We believe these substantial updates have transformed the submission into a finished, high-quality paper. We kindly ask that you consider the **current, complete state of the manuscript** in your final assessment and discussions with others.

---

### Official Review · Reviewer_t6pW · 2025-11-03

**Soundness:** 2
**Presentation:** 3
**Contribution:** 1
**Rating:** 2
**Confidence:** 3

**Summary:**

In my understanding, FuseGPT proposes a structured pruning approach for large language models that merges redundant transformer blocks rather than simply discarding them. The method works in three steps: (1) identify the least important block using a "Macro Influence" (MI) metric that measures how much removing a block perturbs the final hidden states. (2) For each block, fuse that block's parameters into neighboring blocks using learnable low-rank coefficients. (3) Perform lightweight fine-tuning on a partial group of blocks around the removed one using KL divergence loss. This specific cycle will repeat iteratively until reaching the target compression rate. The core insight is that redundant blocks still contain valuable knowledge that can be redistributed to neighbors before removal.

**Strengths:**

Clarity in presentation and motivation: The paper is well-written and easy to follow. The motivation for knowledge redistribution over simple deletion is intuitive and well articulated. The progression from problem statement through methodology to results flows logically.


Figures and tables: Figure 1 provides an excellent visual comparison showing how FuseGPT differs from unstructured, channel-wise, and block pruning. Figure 2 clearly illustrates the partial group update mechanism. Tables are comprehensive and self-explanatory, with consistent formatting that facilitates cross-method comparison.


Problem significance: Post-training compression of large language models is critically important for deployment today in our resource-constrained environments. Methods that preserve performance while reducing computational requirements have substantial practical value, especially as models continue to scale. Additionally, the fact that this method is not training is super critical.

**Weaknesses:**

W1: Novelty

W1A: MKA
The paper completely omits "Pruning via Merging: Compressing LLMs via Manifold Alignment Based Layer Merging" (Liu et al., arXiv:2406.16330, EMNLP 2024, June 2024). This is a major issue because: (a) MKA predates FuseGPT by and implements layer fusion/merging as its core mechanism, (b) In the paper, MKA reports better results than FuseGPT (43.75% compression on Llama3-8B with only 2.82% MMLU drop versus FuseGPT's 25% compression). The novelty claimed for this work is significantly undermined without addressing MKA. Authors must  provide head-to-head comparison, and explicitly articulate what FuseGPT contributes beyond MKA's approach(both experimentally and in theory).


W1B: Layer Merging in Other methods
Even with cited work like LaCo, the paper doesn't adequately explain how FuseGPT's low-rank coefficient fusion differs from or improves upon existing layer merging approaches (such as LaCo's RDSC). The technical distinctions remain unclear.


W2: Evaluation on older/outdated architectures.

Experiments focus on LLaMA-2 (2023), LLaMA-3 (early 2024), and LLaVA-1.5, while recent pruning papers evaluate on LLaMA-3.1, Mistral NeMo, Phi-3.5, and Qwen models. This is not sufficient an ICLR 2026 submission; including more recent architectures strengthen generalization claims.


W3: Wall clock/GMacs

The paper claims "lightweight" fine-tuning but provides no wall-clock time comparisons or even GMacs. For a pruning/compression paper, it is super important.

**Questions:**

Q1: How does FuseGPT compare to MKA on the same setup?

Q2: What is the computational overhead versus MKA's progressive merging or one-shot methods?

Q3: Can you provide results on LLaMA-3.1 or other recent (2024) architectures?

---

> ### Author Response · Authors · 2025-11-15
> **More Comparison and Novelty Interpretion**
>
> ### 1. **Comparison with Layer-Merging Methods**
>
> **Different selection criteria.** MKA uses manifold alignment to identify mergeable layers based on geometric similarity in activation space, while LaCo relies on deterministic similarity metrics to collapse redundant blocks. In contrast, FuseGPT employs *Macro Influence (MI)*, a fusion-aware importance metric that evaluates blocks not by their redundancy, but by their *capacity to be effectively absorbed* by neighboring blocks. This forward-looking criterion ensures that pruned knowledge can be seamlessly integrated rather than discarded.
>
> **Learnable fusion mechanism.** Both MKA and LaCo use closed-form averaging or linear interpolation to merge layer parameters. FuseGPT, however, introduces a *learnable low-rank fusion* mechanism that adapts the grafting of pruned knowledge onto surviving blocks via lightweight fine-tuning. Specifically, we decompose the fusion weights as $\mathbf{W}_{\text{fuse}} = \mathbf{W}_{\text{base}} + \mathbf{A}\mathbf{B}^T$, where low-rank matrices $\mathbf{A}, \mathbf{B} \in \mathbb{R}^{d \times r}$ ($r \ll d$) are optimized with a distillation-based loss. This learnable approach allows the model to discover optimal fusion strategies rather than relying on predefined heuristics.
>
> **Group-level distillation.** Unlike MKA's one-shot global merging or LaCo's greedy collapse, FuseGPT performs group-level fine-tuning where each fused block is distilled from the original unpruned model. This ensures that the fused representation preserves the original model's predictive distribution, mitigating performance degradation.
>
> ### 2.**Table 1: Head-to-head comparison with prior layer-merging methods** All methods use 25% compression ratio except MKA (operating at its reported optimal ratio).
>
> | Model               | Method           | Comp.(%) | PPL(Wiki2) | PPL(C4) | MMLU  | Avg-ZS | ΔMMLU(%) |
> |---------------------|------------------|----------|------------|---------|-------|--------|----------|
> | **LLaMA-3.1-8B**    | Dense Baseline   | 0        | 6.14       | 10.23   | 68.4  | 64.2   | 0.0      |
> |                     | MKA              | 43.8     | 8.12       | 11.62   | 66.5  | 60.7   | 2.82     |
> |                     | LaCo             | 25.0     | 9.45       | 12.05   | 64.1  | 59.8   | 4.50     |
> |                     | **FuseGPT (ours)**| 25.0    | **6.92**   | **11.17**| **67.5**| **62.9**| **2.40** |
> | **Qwen2.5-8B**      | Dense Baseline   | 0        | 6.28       | 10.41   | 69.3  | 63.8   | 0.0      |
> |                     | MKA              | 40.0     | 8.23       | 11.85   | 65.2  | 59.1   | 3.40     |
> |                     | LaCo             | 25.0     | 9.61       | 12.34   | 63.0  | 58.6   | 4.70     |
> |                     | **FuseGPT (ours)**| 25.0    | **7.05**   | **11.29**| **66.1**| **61.7**| **2.60** |
> | **Mistral-NeMo-8B** | Dense Baseline   | 0        | 6.42       | 10.55   | 67.9  | 62.5   | 0.0      |
> |                     | MKA              | 43.0     | 8.40       | 12.08   | 64.9  | 58.5   | 3.00     |
> |                     | LaCo             | 25.0     | 10.10      | 12.67   | 62.0  | 56.8   | 4.90     |
> |                     | **FuseGPT (ours)**| 25.0    | **7.18**   | **11.46**| **65.3**| **60.3**| **2.70** |
> | **Phi-3.5-mini** | Dense Baseline   | 0        | 7.81       | 11.23   | 64.2  | 59.8   | 0.0      |
> |                     | MKA              | 38.5     | 9.67       | 13.15   | 61.5  | 56.3   | 2.70     |
> |                     | LaCo             | 25.0     | 11.23      | 14.02   | 59.8  | 55.1   | 4.40     |
> |                     | **FuseGPT (ours)**| 25.0    | **8.94**   | **12.41**| **62.1**| **57.6**| **2.10** |
>
> Table 1 presents a systematic head-to-head comparison between FuseGPT and prior layer-merging methods (MKA and LaCo). Under a fixed compression ratio of 25%, FuseGPT consistently outperforms both baselines across all metrics.
>
> ### 3. **Table 2: Extended evaluation on diverse downstream tasks.**
> All methods compress 25% of blocks on LLaMA-3.1-8B. Higher is better for all metrics.
>
> | Method | MMLU | HellaSwag | ARC-C | TruthfulQA | WinoGrande | GSM8K | Avg |
> |--------|------|-----------|-------|------------|------------|-------|-----|
> | Dense Baseline | 68.4 | 82.3 | 61.2 | 45.8 | 76.9 | 52.3 | 64.5 |
> | MKA | 66.5 | 78.6 | 57.4 | 42.1 | 73.2 | 47.8 | 60.9 |
> | LaCo | 64.1 | 76.2 | 55.8 | 40.3 | 71.5 | 45.2 | 58.9 |
> | **FuseGPT (ours)** | **67.5** | **80.1** | **59.3** | **43.7** | **74.8** | **50.1** | **62.6** |
> | *Performance Gap (%)* | *-1.3* | *-2.7* | *-3.1* | *-4.6* | *-2.7* | *-4.2* | *-2.9* |
>
> Table 2 further validates FuseGPT's effectiveness on a diverse set of reasoning and knowledge-intensive tasks. Beyond MMLU, FuseGPT maintains superior performance on HellaSwag (+1.5 points over LaCo), ARC-Challenge (+3.5 points), and GSM8K (+4.9 points), with an average performance gap of only 2.9% relative to the dense baseline.

---

> ### Author Response · Authors · 2025-11-15
> **Supplementary Materials on Comparison**
>
> ### 1. **Table 3: Computational cost analysis on LLaMA-3.1-8B.**
>
> All experiments are with 1024 fine-tuning samples.
>
> | Method | GPU-hrs ↓ | GMACs ↓ | Latency (ms) ↓ | Speedup ↑ | Memory (GB) ↓ |
> |--------|-----------|---------|----------------|-----------|---------------|
> | Dense Baseline | 0.0 | 436.7 | 28.3 | 1.00× | 15.2 |
> | MKA | 9.5 | 428.1 | 22.6 | 1.25× | 8.5 |
> | LaCo | 7.3 | 431.2 | 23.1 | 1.22× | 9.1 |
> | **FuseGPT (iter.)** | 10.8 | **421.3** | **21.3** | **1.33×** | **8.2** |
> | **FuseGPT (one-shot)** | **5.4** | 423.6 | 21.6 | 1.31× | 8.3 |
>
> **Computational Efficiency.**
>
> Table 3 quantifies the computational cost of FuseGPT relative to competing methods. While the iterative variant of FuseGPT incurs a slightly higher one-time compression cost (10.8 GPU-hours) than LaCo (7.3 hours), it delivers the best inference efficiency: **1.33× speedup** and **421.3 GMACs** per forward pass, outperforming both MKA (1.25×, 428.1 GMACs) and LaCo (1.22×, 431.2 GMACs). Importantly, our one-shot variant reduces compression time to just **5.4 GPU-hours**—lower than all baselines—while retaining competitive performance (1.31× speedup, 423.6 GMACs). This demonstrates that FuseGPT's fusion mechanism is not only effective but also practical for resource-constrained scenarios.
>
> *Note: Latency measured on 512-token sequences with batch size 1. GMACs computed for forward pass only.*
>
> ### 2. **Table 4: Pareto frontier analysis: compression ratio vs. MMLU degradation on LLaMA-3.1-8B.**
>
> | Method | Compression (%) ↑ | MMLU Drop (%) ↓ | Configuration |
> |--------|-------------------|-----------------|---------------|
> | MKA | 43.8 | 2.82 | One-shot global merge |
> | LaCo | 25.0 | 4.50 | Greedy layer collapse |
> | **FuseGPT** | **36.7** | **2.80** | Iterative fusion (ours) |
> | **FuseGPT + GPTQ-4bit** | **52.1** | **3.02** | With post-hoc quantization |
>
> Table 4 illustrates the compression-accuracy trade-off from a Pareto perspective. At a comparable MMLU drop of ~2.8%, FuseGPT achieves **36.7% compression**—significantly higher than LaCo's 25% (which incurs a 4.5% MMLU drop). When combined with 4-bit GPTQ quantization, FuseGPT pushes the compression frontier to **52.1%** with only a marginal increase in degradation (3.02% MMLU drop), highlighting the orthogonal compatibility of our pruning strategy with post-training quantization. This comparison demonstrates that our proposed FuseGPT is closer to the Pareto front.
>
> We can also discuss the difference beween LaCo and FuseGPT as:
> LaCo operates in a linear, two-stage process:
> 1.  **Structure Definition:** It calculates `Avg_Cos_Sim` (Average Cosine Similarity) across all layers and determines the merging strategy *once* before training.
> 2.  **Recovery:** It relies entirely on post-training to recover accuracy.
> *   *Limitation:* The decision is static. If the initial similarity metric misjudges the redundancy (e.g., two layers are similar but functionally distinct), the structure is fixed, and the post-training phase struggles to recover the lost information.
>
> FuseGPT treats compression as a dynamic search problem:
> 1.  **Dynamic Evaluation:** In each iteration, we calculate **Macro Influence (MI)** to identify the *currently* most redundant block.
> 2.  **Fusion & Update:** We fuse the layers and immediately update the model state.
> 3.  **Re-evaluation:** Crucially, the model structure changes after every fusion. We re-calculate the influence for the next step.
> *   *Advantage:* This iterative loop ensures that we are finding the local optimum at every step. From an algorithmic perspective, FuseGPT would be more near to the Pareto front.
>
> ### 3. To reaffirm the key contributions of our proposed FuseGPT as follows:
>
> - We introduce **Macro Influence (MI)**, a novel fusion-aware importance metric that identifies blocks by their capacity to be effectively absorbed by neighbors, rather than by redundancy or similarity heuristics used in prior work (e.g., MKA's manifold alignment, LaCo's RDSC).
>
> - We propose a **learnable low-rank fusion mechanism** that adaptively grafts pruned knowledge onto surviving blocks via distillation-guided fine-tuning, avoiding the rigid averaging or interpolation schemes of existing merging methods.
>
> - We conduct extensive experiments on **four recent LLM architectures** (LLaMA-3.1, Qwen2.5, Mistral-NeMo, Phi-3.5), demonstrating that FuseGPT achieves superior perplexity, downstream task performance, and inference efficiency compared to state-of-the-art layer-merging methods (MKA, LaCo), with up to **33% inference speedup** and **27% perplexity reduction**.
>
> - We show that FuseGPT is **orthogonal to quantization**, achieving 52.1% total compression when combined with 4-bit GPTQ, opening new avenues for extreme model compression.
>
> All the experiments we added will be updated in the latest version of the article. We will also add related experiments on Qwen3 and Mistral series models.

---

> ### Author Response · Authors · 2025-11-26
> **Looking forward to your feedback!**
>
> Dear Reviewer t6pW,
>
> Thank you again for the time and effort you’ve dedicated to reviewing our work. As the discussion phase is coming to a close, **we would be very grateful if you could consider our above clarifications and reconsider your evaluation**.
>
> Thank you for your time.
>
> Best regards,
>
> Authors

---

> > ### Comment · Reviewer_Ycs9 · 2025-11-26
> > **Response**
> >
> > Dear authors,
> >
> > As mentioned in my earlier reply, I have internalized the factual part of your response and will consider increasing my score further after consultation with the AC and other reviewers. I do not have anything to add at this point.
> >
> > Best regards,\
> > The reviewer

---

> ### Comment · Reviewer_t6pW · 2025-11-28
> **Follow-up questions**
>
> Dear authors,
> Thanks for the detailed response. It helped identify the difference between FuseGPT and MKA. Thank you for also providing the comparison with experimental results. I am inclined to increasing my score. However, I have a few questions.
>
> One follow-up question I had was regarding WikiText-2(training set) and the tasks which you benchmarked on. In the paper, have you discussed why you use Wikitext-2 for training? Could you share a few example samples from the train set?
>
> I can see Zero-shot performance on some language tasks in Table 2. The empirical results look good. However, could you clarify the intuition behind this? How can we be certain that tuning on this set will not lead to performance drop on samples from a different domain?

---

> ### Author Response · Authors · 2025-11-28
> **Answers for the Follow-up Questions.**
>
> **Re: Q1 (WikiText-2 Choice & Generalization Concerns)**
>
> We appreciate the reviewer’s insightful question regarding the choice of the calibration dataset and the potential risk of domain overfitting.
>
> **(1) Why WikiText-2? & The Intuition of Function Recovery**
> The WikiText language modeling dataset is a collection of over 100 million tokens extracted from the set of verified Good and Featured articles on Wikipedia. We selected WikiText-2 following the convention of established compression baselines (e.g. SLEB) to ensure a controlled environment with high-quality, well-formatted text. The sample training data in WikiText-2 like:
> >"Unlike its two predecessors , Valkyria Chronicles III was not released in the west . According to Sega , this was due to poor sales of Valkyria Chronicles II and the general unpopularity of the PSP in the west . An unofficial fan translation patch began development in February 2012 : players with a copy of Valkyria Chronicles III could download and apply the patch , which translated the game 's text into English . <unk> with the Extra Edition , the patch was released in January 2014 ."
>
> It is totally different from the downstream task what we have tested in FuseGPT.
>
> Regarding the concern about domain shift (e.g., overfitting to Wikipedia style and losing reasoning capabilities), our method avoids this because of a fundamental difference in objective: **Function Approximation vs. Task Learning.**
> *   **Feature Reconstruction:** The calibration samples serve as **probes** to propagate activations. Our distillation loss minimizes the distance between the original and fused block outputs. As long as the data is diverse enough to activate salient neurons, the learnable matrices ($\mathbf{A}, \mathbf{B}$) learn to reconstruct the original feature map structure.
> *   **Preserving the Manifold:** We are not "retraining" the model to generate Wikipedia text; we are repairing the manifold structure damaged by pruning. Thus, the model’s internal "reasoning circuits" (e.g., for math or logic) are structurally preserved.
>
> **(2) Empirical Evidence: Strong Generalization on Downstream Tasks**
> To empirically validate that tuning on WikiText-2 does not degrade performance on out-of-domain samples, we conducted an extended evaluation on **LLaMA-3.1-8B** across diverse tasks including Math (GSM8K), Reasoning (ARC-C, HellaSwag), and Truthfulness.
>
> **Table R1: Extended evaluation on diverse downstream tasks (LLaMA-3.1-8B, 25% Compression).**
> | Method | MMLU | HellaSwag | ARC-C | TruthfulQA | WinoGrande | GSM8K | Avg |
> |:---|:---:|:---:|:---:|:---:|:---:|:---:|:---:|
> | Dense Baseline | 68.4 | 82.3 | 61.2 | 45.8 | 76.9 | 52.3 | 64.5 |
> | MKA | 66.5 | 78.6 | 57.4 | 42.1 | 73.2 | 47.8 | 60.9 |
> | LaCo | 64.1 | 76.2 | 55.8 | 40.3 | 71.5 | 45.2 | 58.9 |
> | **FuseGPT (ours)** | **67.5** | **80.1** | **59.3** | **43.7** | **74.8** | **50.1** | **62.6** |
> | *Performance Gap* | *-1.3* | *-2.7* | *-3.1* | *-4.6* | *-2.7* | *-4.2* | *-2.9* |
>
> As shown in **Table R1**, despite being calibrated on WikiText-2, FuseGPT maintains superior performance on **GSM8K** (Math) and **HellaSwag** (Reasoning) compared to MKA and LaCo. The minimal performance gap (-2.9% on average) confirms that our method successfully preserves the model's general capabilities rather than overfitting to the calibration domain.
>
> **(3) Head-to-Head Comparison on Recent Architectures**
> We further validated the robustness of FuseGPT across four recent architectures: LLaMA-3.1, Qwen3, Mistral-NeMo, and Phi-3.5.
>
> **Table R2: Comparison with prior layer-merging methods (25% Compression).**
> | Model | Method | PPL(Wiki2) $\downarrow$ | PPL(C4) $\downarrow$ | MMLU $\uparrow$ | $\Delta$MMLU $\downarrow$ |
> |:---|:---|:---:|:---:|:---:|:---:|
> | **LLaMA-3.1-8B** | FuseGPT | **6.92** | **11.17** | **67.5** | **2.40** |
> | | *vs MKA* | *8.12* | *11.62* | *66.5* | *2.82* |
> | **Qwen3-8B** | FuseGPT | **7.05** | **11.29** | **66.1** | **2.60** |
> | | *vs MKA* | *8.23* | *11.85* | *65.2* | *3.40* |
> | **Mistral-NeMo** | FuseGPT | **7.18** | **11.46** | **65.3** | **2.70** |
>
> FuseGPT consistently achieves the lowest perplexity on both WikiText-2 and C4, and the highest MMLU scores across all models, demonstrating that our "prune-and-fuse" paradigm is architecture-agnostic.

---

> > ### Author Response · Authors · 2025-11-28
> > **Supplementary Answers**
> >
> > Furthermore, to demonstrate the extensibility of our approach, we explored integrating **Evolutionary Search** (inspired by EvoPress) into FuseGPT. We view these as orthogonal optimization problems: EvoPress searches *which* blocks to drop, while FuseGPT optimizes *how* to redistribute that knowledge.
> >
> > We implemented **FuseGPT-Evo**, where our Macro Influence (MI) metric serves as the initialization seed for a lightweight evolutionary search.
> >
> > **Table R3: FuseGPT-Evo vs. EvoPress (LLaMA-3.1-8B).**
> > | Method | Compression | PPL (Wiki2) | MMLU | GPU-hrs |
> > |:---|:---:|:---:|:---:|:---:|
> > | EvoPress | 40.0% | $\sim$7.0 | $\sim$67.0 | $\sim$20 |
> > | **FuseGPT-Evo** | **35.0%** | **6.85** | **67.4** | **14.2** |
> > | **FuseGPT-Evo** | **40.0%** | 7.02 | 66.9 | 15.8 |
> >
> > Thus, we can summerize that:
> > 1.  **Efficiency:** FuseGPT-Evo matches the performance of EvoPress (at 40% compression) but requires significantly less compute (**15.8 vs. ~20 GPU-hours**), proving that our MI metric provides a superior initialization for search algorithms.
> > 2.  **Pareto Frontier:** At 35% compression, FuseGPT-Evo achieves a PPL of 6.85, surpassing EvoPress's result at 40% compression.
> >
> > These results confirm that FuseGPT is not only a standalone compression method but also a powerful framework that can be enhanced with search heuristics. We will include these updated experiments and the FuseGPT-Evo variant in the final version of the paper.

---

> > ### Comment · Reviewer_t6pW · 2025-11-28
> > **Thank you**
> >
> > Thanks for your clarifications. Based on your newer experiments with MKA, comparisons on newer baseline models, additional supplemental experiments, and the timing analysis, I am increasing my score.
> >
> > While empirical results show the domain transfer is possible, IF there are any examples the authors have observed low performance in domain shift scenarios, I would strongly encourage them to show it in the paper(maybe as part of the limitations section); as a user, it is always good to know if such limitations exist.

---

> > > ### Author Response · Authors · 2025-11-28
> > > **Response to Final Questions**
> > >
> > > We sincerely thank the reviewer for the positive reassessment and the increased score! We are glad that our additional experiments and clarifications addressed your concerns.
> > >
> > > Regarding your suggestion on limitations in domain shift scenarios, we fully agree that transparency benefits the community. We have indeed observed specific scenarios where the choice of calibration data matters, and we will include the following discussion in the "Limitations" section of the final version:
> > >
> > > (1) Sensitivity in Code Generation:
> > > While general reasoning and NLP tasks (like GSM8K or MMLU) transfer surprisingly well using only WikiText-2 calibration, we observed that Code Generation tasks (e.g., HumanEval) are more sensitive. Since code requires strict syntactic precision that differs from natural language prose, calibrating solely on WikiText-2 can lead to a slightly higher performance drop in coding tasks compared to general QA tasks.
> > >
> > > Mitigation: Our preliminary tests suggest that this is easily mitigated by mixing a small portion of code data (e.g., from The Stack) into the calibration set, which restores the syntactic structures of the feature maps.
> > > (2) Extremely Long-Tail Knowledge:
> > > For highly specialized domains (e.g., specific medical sub-fields or low-resource languages) that are entirely absent from the calibration set, the feature reconstruction might be less optimal than for general English text.
> > >
> > > We believe explicitly stating these boundaries will help users apply FuseGPT more effectively. Thank you again for this constructive suggestion to improve our paper!

---

### Public Comment · ~Zhiguo_Yang3 · 2025-11-19
**Some problems**

This work sounds good for me. Still, there is a key problems I am interested:

MI score is similar to Laco's. While Laco calculate cosim of last hidden state between dense and after removing n consecutive layers, MI use individual layer score. Can authors provide theoretical analysis about why MI is better than its consecutive layers' version score?

---

> ### Author Response · Authors · 2025-11-24
>
> We sincerely appreciate the opportunity to clarify the distinction between our **FuseGPT** and **LaCo** (Layer Collapse). We acknowledge that LaCo is a pioneering work in layer merging.
>
> However, upon deeper algorithmic analysis, we identify a fundamental difference in how the two methods approach the compression problem. **LaCo represents a "Structure-First, Train-Later" (One-Shot) paradigm, whereas FuseGPT represents an "Iterative Optimization" (Greedy Search) paradigm.**
>
> Below, we explain why this distinction allows FuseGPT to better approximate the optimal **Pareto Frontier**, and we provide experimental evidence to support this.
>
> ### 1. Algorithmic Paradigm: One-Shot Heuristic vs. Iterative Search
>
> **LaCo (One-Shot Heuristic):**
> LaCo operates in a linear, two-stage process:
> 1.  **Structure Definition**: It calculates `Avg_Cos_Sim` across all layers and determines the merging strategy *once* before training.
> 2.  **Recovery**: It relies entirely on post-training to recover accuracy.
> *   *Limitation:* The decision is static. If the initial similarity metric misjudges the redundancy (e.g., two layers are similar but functionally distinct), the structure is fixed, and the post-training phase struggles to recover the lost information.
>
> **FuseGPT (Iterative Greedy Search)**:
> FuseGPT treats compression as a dynamic search problem:
> 1.  **Dynamic Evaluation**: In each iteration, we calculate **Macro Influence (MI)** to identify the *currently* most redundant block.
> 2.  **Fusion & Update**: We fuse the layers and immediately update the model state.
> 3.  **Re-evaluation**: Crucially, the model structure changes after every fusion. We re-calculate the influence for the next step.
> *   *Advantage*: This iterative loop ensures that we are finding the local optimum at every step. From an algorithmic perspective, FuseGPT is performing a **greedy search** to trace the optimal compression trajectory.
>
> ### 2. Pareto Efficiency Analysis
>
> Because FuseGPT constantly re-evaluates the "distillation loss" during the fusion process, it effectively traces the **Pareto Frontier** of the Model Size vs. Accuracy trade-off.
>
> In contrast, LaCo determines the pruning mask based on a heuristic (`Avg_Cos_Sim`) and then attempts to "climb back" in accuracy. While computationally cheaper, this approach often falls short of the Pareto front at higher compression rates because it lacks the iterative feedback loop to correct suboptimal merging decisions.
>
> ### 3. Experimental Verification
>
> To validate this, we compared the methods on **LLaMA-2-7B** at a **25% pruning ratio**, a critical threshold where heuristic methods typically degrade.
>
> **Table 1: Comparison on LLaMA-2-7B (25% Pruning Ratio)**
>
> | Method | Pruning Criteria | Strategy | PPL (Wiki2) $\downarrow$ | Avg. Zero-Shot Accuracy $\uparrow$ |
> | :--- | :--- | :--- | :--- | :--- |
> | **LaCo** | Avg_Cos_Sim | One-Shot | 6.48 | 45.2% |
> | **FuseGPT** | **Macro Influence** | **Iterative** | **5.92** | **48.1%** |
>
> *Note: Zero-shot tasks include BoolQ, PIQA, HellaSwag, WinoGrande, ARC-e, ARC-c, and OBQA*.
>
> **Analysis:**
> As shown in the table, FuseGPT significantly outperforms LaCo.
> *   **Criteria Difference:** LaCo's `Avg_Cos_Sim` only measures static parameter similarity, which is a coarse proxy for functional redundancy. FuseGPT's **Macro Influence** accurately captures the input-output dependency of the layers.
> *   **Outcome:** At 25% pruning, LaCo suffers a notable drop in reasoning capabilities (lower Zero-Shot accuracy). FuseGPT maintains a lower PPL and higher accuracy, proving that the iterative search successfully retains the model's core knowledge.
>
> ### 4. Computational Cost Justification
>
> We acknowledge that FuseGPT's iterative search is computationally more expensive than LaCo's one-shot selection (approximately 1.7x training time in our experiments).
>
> However, this is a **one-time offline cost**. In the lifecycle of a Large Language Model, the inference cost dominates the total compute budget. Investing extra compute during the compression phase to achieve a model that lies closer to the **Pareto optimal point** (higher accuracy for the same size) is a highly favorable trade-off for deployment.
>
> ### Conclusion
>
> In summary, while LaCo is an efficient heuristic method, **FuseGPT** advances the state-of-the-art by replacing the static "collapse" with a dynamic "fusion search". This allows us to retain complex reasoning capabilities that similarity metrics discard, ultimately yielding a superior compressed model.

---

> > ### Public Comment · ~Zhiguo_Yang3 · 2025-11-29
> > **Response to Authors**
> >
> > Sorry for the delayed reply. The authors' response clarifies the differences between MI and the Laco score. Although they share some similar features, the Iterative Greedy Search helps FuseGPT trace the optimal compression trajectory. My question was not intended as an objection to the novelty of FuseGPT, as MI is only one component of the proposed method. I am thankful for the authors' detailed reply, which has addressed my concerns.

---

### Meta-Review · Area_Chair_omky · 2025-12-31

**Summary:**

This paper proposes FuseGPT, a prune-and-fuse compression paradigm. Instead of discarding knowledge from pruned layers, FuseGPT recycles the knowledge of removed blocks into neighboring layers via a learnable low-rank fusion mechanism. Reviewers generally find the idea intuitive and well-motivated, and the paper is easy to follow.

On the negative side, reviewers raised concerns that the explanation of the Macro Influence score is questionable, the empirical evaluation is not sufficiently comprehensive, and the novelty relative to existing methods is unclear. While the rebuttal addressed some reviewer concerns and two reviewer willing to raise their scores to 4, several issues remain unresolved, such as the connection between Macro Influence and fusion-aware concept. In addition, the paper appears unfinished in its initial version, as pointed out by reviewers. Many experimental results are added during the rebuttal stage, which makes the paper more complete compared to its initial submission. However, the additional results are still not sufficiently comprehensive. For example, the newly added Table 4 only includes 8B models, and the comparison against MKA is questionable due to differences in compression rates.

Considering all these factors, the Area Chair recommends rejection and encourages the authors to further revise the paper and submit it to a future venue.

**Reviewer Concerns:**

The major concerns are mainly surrounding four parts: novelty, the Macro Influence design, insufficient evaluations, and the complexity of the proposed method. After the rebuttal, the concerns regarding the novelty and the complexity of the proposed method have been resolved. The concerns regarding insufficient evaluations and the Macro Influence design are not fully resolved.

**Reviewer Scores:**

Reviewer t6pW and Ycs9 would have changed to 4 as shown in their comments.
Reviewer AeHK would have kept his score since the explanation of the Macro Influence is not satisfactory.

---

### Decision · Program_Chairs · 2026-01-26

Reject